# A Framework for Robustness Certification of Smoothed Classifiers using f-Divergences

**Krishnamurthy (Dj) Dvijotham**[†]   **Jamie Hayes**[†‡*]   **Borja Balle**[†]   **J. Zico Kolter**[ac]
**Chongli Qin**[†]   **András György**[†]   **Kai Xiao**[b†*]   **Sven Gowal**[†]   **Pushmeet Kohli**[†]
[†]DeepMind, London, UK   [‡]University College London, UK
[a] Carnegie Mellon University, USA   [b] Massachusetts Institute of Technology, USA
[c] Bosch Center for AI, USA
{dvij, bballe, chongliqin, agyorgy, sgowal, pushmeet}@google.com
{zkolter}@cs.cmu.edu  {j.hayes}@ucl.ac.uk  {kaix}@mit.edu

## Abstract

Formal verification techniques that compute provable guarantees on properties of machine learning models, like robustness to norm-bounded adversarial perturbations, have yielded impressive results. Although most techniques developed so far require knowledge of the architecture of the machine learning model and remain hard to scale to complex prediction pipelines, the method of *randomized smoothing* has been shown to overcome many of these obstacles. By requiring only black-box access to the underlying model, randomized smoothing scales to large architectures and is agnostic to the internals of the network. However, past work on randomized smoothing has focused on restricted classes of smoothing measures or perturbations (like Gaussian or discrete) and has only been able to prove robustness with respect to simple norm bounds. In this paper we introduce a general framework for proving robustness properties of smoothed machine learning models in the black-box setting. Specifically, we extend randomized smoothing procedures to handle *arbitrary* smoothing measures and prove robustness of the smoothed classifier by using $f$-divergences. Our methodology improves upon the state of the art in terms of computation time or certified robustness on several image classification tasks and an audio classification task, with respect to several classes of adversarial perturbations.

## 1 Introduction

Predictors obtained from machine learning algorithms have been shown to be vulnerable to making errors when the inputs are perturbed by carefully chosen small but imperceptible amounts (Szegedy et al., 2014; Biggio et al., 2013). This has motivated significant amount of research in improving adversarial robustness of a machine learning model (see, e.g. Goodfellow et al., 2015; Madry et al., 2018). While significant advances have been made, it has been shown that models that were estimated to be robust have later been broken by stronger attacks (Athalye et al., 2018; Uesato et al., 2018). This has led to the need for methods that offer provable guarantees that the predictor cannot be forced to misclassify an example by *any attack algorithm* restricted to produce perturbations within a certain set (for example, within an $\ell_p$ norm ball). While progress has been made leading to methods that are able to compute provable guarantees for several image and text classification tasks (Wong & Kolter, 2018; Wong et al., 2018; Raghunathan et al., 2018; Dvijotham et al., 2018; Katz et al., 2017; Huang et al., 2019; Jia et al., 2019), these methods require extensive knowledge of the architecture of the predictor and are not easy to extend to new models or architectures, requiring specialized algorithms for each new class of models. Furthermore, the computational complexity of these methods grows significantly with input dimension and model size.

To deal with these obstacles, recent work has proposed the *randomized smoothing* strategy for verifying the robustness of classifiers. Specifically, Lecuyer et al. (2019) and Cohen et al. (2019)

---

*Work done during an internship at DeepMind.

have shown that robustness properties can be more easily verified for the *smoothed* version of a base classifier $h$ producing labels in some set $\mathcal{Y}$:

$$h_s(x) = \arg\max_{y \in \mathcal{Y}} \mathbb{P}_{X \sim \mu(x)}[h(X) = y] \; , \tag{1}$$

where the labels returned by the smoothed classifier $h_s$ are obtained by taking a "majority vote" over the predictions of the original classifier $h$ on random inputs drawn from a probability distribution $\mu(x)$, called the *smoothing measure*. Lecuyer et al. (2019) showed that verifying the robustness of this smoothed classifier is significantly simpler than verifying the original classifier $h$ and only requires estimating the distribution of outputs of the classifier under random perturbations of the input, but does not require access to the internals of the classifier $h$. We refer to this as *black-box verification*.

In this work, we develop a general framework for black-box verification that recovers prior work as special cases, and improves upon previous results in various ways.

**Contributions**   Our contributions are summarized as follows:

1. We formulate the general problem of black-box verification via a generalized randomized smoothing procedure, which extends existing approaches to allow for arbitrary smoothing measures. Specifically, we show that robustness certificates for smoothed classifiers can be obtained by solving a small convex optimization problem when allowed adversarial perturbations can be characterized via divergence-based bounds on the smoothing measure.

2. We prove that our certificates generalize previous results obtained in related work (Lecuyer et al., 2019; Cohen et al., 2019; Li et al., 2019), and vastly extend the class of perturbations and smoothing measures that can be used while still allowing certifiable guarantees.

3. We introduce the notion of full-information and information-limited settings, and show that the information-limited setting that has been the main focus of prior work leads to weaker certificates for smoothed probabilistic classifiers, and can be improved by using additional information (the distribution of label scores under randomized smoothing).

4. We evaluate our framework experimentally on image and classification tasks, obtaining robustness certificates that improve upon other black-box methods either in terms of certificate tightness or computation time on robustness to $\ell_0, \ell_1$ or $\ell_2$ perturbations on MNIST, CIFAR-10 and ImageNet. $\ell_2$ perturbations result from worst-case realizations of white noise that is common in many image, speech and video processing. $\ell_0$ perturbations can model missing data (missing pixels in an image, or samples in a time-domain audio signal) while $\ell_1$ perturbations can be used to model convex combinations of discrete perturbations in text classification (Jia et al., 2019). We also obtain the first, to the best of our knowledge, certifiably robust model for an audio classification task, Librispeech (Panayotov et al., 2015), with variable-length inputs.

## 2   BLACK-BOX VERIFICATION FOR SMOOTHED CLASSIFIERS

Consider a binary classifier $h : \mathcal{X} \to \{\pm 1\}$ given to us as a black box, so we can only access the inputs and outputs of $h$ but not its internals. We are interested in investigating the robustness of the smoothed classifier $h_s$ (defined in Eq. 1) against adversarial perturbations of size at most $\epsilon$ with respect to a given norm $\|\cdot\|$. To determine whether a norm-bounded adversarial attack on a fixed input $x \in \mathcal{X}$ with $h_s(x) = +1$ could be successful, we can solve the optimization problem

$$\min_{\|x'-x\| \leq \epsilon} \mathbb{P}_{X' \sim \mu(x')}[h(X') = +1] \; , \tag{2}$$

and check whether the minimum value can be smaller than $\frac{1}{2}$. This is a non-convex optimization problem for which we may not even be able to compute gradients since we only have black-box access to $h$. While techniques have been developed to address this problem, obtaining provable guarantees on whether these algorithms actually find the worst-case adversarial perturbation is difficult since we do not know anything about the nature of $h$.

Motivated by this difficulty, we take a different approach: Rather than studying the adversarial attack in the input space $\mathcal{X}$, we study it in the space of probability measures over inputs, denoted by $\mathcal{P}(\mathcal{X})$.

Formally, this amounts to rewriting Eq. 2 as

$$\min_{\nu \in \{\mu(x') : \|x' - x\| \leq \epsilon\}} \mathbb{P}_{X' \sim \nu}[h(X') = +1] \ . \tag{3}$$

This is an infinite dimensional optimization problem over the space of probability measures $\nu \in \mathcal{P}(\mathcal{X})$ subject to the constraint $\nu \in \mathcal{D} = \{\mu(x') : \|x' - x\| \leq \epsilon\}$. While this set is still intractable to deal with, we can consider relaxations of this set defined by divergence constraints between $\nu$ and $\rho = \mu(x)$, i.e., $\mathcal{D} \subseteq \{\nu : D(\nu \| \rho) \leq \epsilon_D\}$ where $D$ denotes some divergence between probability distributions. We will show in Section 3 that for several commonly used divergences (in fact, for any $f$-divergence; cf. Ali & Silvey, 1966), the relaxed problem can be solved efficiently.

## 2.1 A GENERAL FRAMEWORK FOR ROBUSTNESS CERTIFICATION

To formulate the general verification problem, consider a *specification* $\phi : \mathcal{X} \to \mathcal{Z} \subseteq \mathbb{R}$: a generic function over the input space (that typically is a function of the classifier output) that we want to verify has certain properties. Unless otherwise specified, we will assume that $\mathcal{X} \subseteq \mathbb{R}^d$ (we work in a $d$ dimensional input space). Our framework also involves a reference measure $\rho$ (in the above example we would take $\rho = \mu(x)$) and a collection of perturbed distributions $\mathcal{D}$ (in the above example we would take $\mathcal{D} = \mathcal{D}_{x,\epsilon} = \{\mu(x') : \|x' - x\| \leq \epsilon\}$).

Verifying that a given specification $\phi$ is robustly certified is equivalent to checking whether the optimal value of the optimization problem

$$\text{OPT}(\phi, \rho, \mathcal{D}) := \min_{\nu \in \mathcal{D}} \mathrm{E}_{X \sim \nu}[\phi(X)] \ , \tag{4}$$

is non-negative. Solving problems of this form is the key workhorse of our general framework for black-box certification of adversarial robustness for smoothed classifiers.

Using these ingredients we introduce two closely related certification problems: *information-limited robust certification* and *full-information robust certification*. In the former case, we assume that we are given only given access to $\mathbb{P}_{X \sim \rho}[\phi(X) = +1], \mathbb{P}_{X \sim \nu}[\phi(X) = +1]$. In the latter case, we are given full-access to specification $\phi$. The definitions are below.

**Definition 2.1** (Information-limited robust certification). Given *reference distribution* $\rho \in \mathcal{P}(\mathcal{X})$, probabilities $\theta_a, \theta_b$ that satisfy $\theta_a, \theta_b \geq 0, \theta_a + \theta_b \leq 1$ and collection of *perturbed distributions* $\mathcal{D} \subset \mathcal{P}(\mathcal{X})$ containing $\rho$, define the class of specifications $S$ as

$$S = \left\{ \phi : \mathcal{X} \to \{-1, 0, +1\} \text{ s.t. } \mathbb{P}_{X \sim \rho}[\phi(X) = +1] \geq \theta_a, \mathbb{P}_{X \sim \rho}[\phi(X) = -1] \leq \theta_b \right\}$$

We say that $S$ is *information-limited robustly certified* at $\rho$ with respect to $\mathcal{D}$ if the following condition holds: $\mathrm{E}_{X \sim \nu}[\phi(X)] \geq 0$ for all $\nu \in \mathcal{D}, \phi \in S$.

Note since we don't have access to $\phi$, we need to prove that $\mathrm{E}_{X \sim \nu}[\phi(X)] \geq 0 \quad \forall \nu \in \mathcal{D}$ is satisfied for all specifications in set $S$. Although the information-limited case may seem challenging because we need to provide guarantees that hold simultaneously over a whole class of specifications, it turns out that, for perturbation sets $\mathcal{D}$ specified by an $f$-divergence bound, this certification task can be solved efficiently using convex optimization.

**Definition 2.2** (Full-information robust certification). Given a *reference distribution* $\rho \in \mathcal{P}(\mathcal{X})$, a *specification* $\phi : \mathcal{X} \to \mathcal{Z} \subseteq \mathbb{R}$ and a collection of *perturbed distributions* $\mathcal{D} \subset \mathcal{P}(\mathcal{X})$ containing $\rho$, we say that $\phi$ is *full-information robustly certified* at $\rho$ with respect to $\mathcal{D}$ if the following condition holds: $\mathrm{E}_{X \sim \nu}[\phi(X)] \geq 0$ for all $\nu \in \mathcal{D}$.

Most often we are dealing with the case where we have full access to the specification $\phi$, thus we should be able to certify using *full-information robust certification*. However, prior works, Cohen et al. (2019) and Lecuyer et al. (2019), have only provided solutions to certify with respect to the *information-limited* case where we cannot use all of the information about $\phi$. The framework we develop is a more general method that can be used in both *information-limited* and *full-information* scenarios. We will demonstrate that our framework recovers certificates provided by Cohen et al. (2019), Li et al. (2019) and dominates Lecuyer et al. (2019) in the information-limited setting (see section 5). Further, it can utilize full-information about the specification $\phi$ to provide tighter certificates for smoothed probabilistic classifiers (see section 6).

### 2.1.1 ROBUSTNESS SPECIFICATION FOR SMOOTHED HARD CLASSIFIERS

We first note that the definitions above are sufficient to capture the standard usage of randomized smoothing as it has been used in past work (e.g. Lecuyer et al., 2019; Cohen et al., 2019) to verify the robustness of smoothed multi-class classifiers. Specifically, consider smoothing a classifier $h : \mathcal{X} \to \mathcal{Y}$ with a finite set of labels $\mathcal{Y}$ using a smoothing measure $\mu : \mathcal{X} \mapsto \mathcal{P}(\mathcal{X})$. The resulting randomly smoothed classifier $h_s$ is defined in Eq. 1. Our goal is to certify that the prediction $h_s(x)$ is robust to perturbations of size at most $\epsilon$ measured by distance function[1] $d : \mathcal{X} \times \mathcal{X} \mapsto \mathbb{R}_+$, i.e.,

$$h_s(x') = h_s(x) \quad \forall x' \text{ such that } d(x, x') \leq \epsilon \ . \tag{5}$$

To pose this question within our framework, we choose the reference distribution $\rho = \mu(x)$, the set of perturbed distributions $\mathcal{D}_{x,\epsilon} = \{\mu(x') : d(x, x') \leq \epsilon\}$, and the following specifications. Let $c = h_s(x)$. For every $c' \in \mathcal{Y} \setminus \{c\}$, we define the specification $\phi_{c,c'} : \mathcal{X} \mapsto \{-1, 0, +1\}$ as follows:

$$\phi_{c,c'}(x) = \begin{cases} +1 & \text{if } h(x) = c \ , \\ -1 & \text{if } h(x) = c' \ , \\ 0 & \text{otherwise} \ . \end{cases}$$

Then, Eq. 5 holds if and only if every $\phi_{c,c'}$, $c' \neq c$, is robustly certified at $\mu(x)$ with respect to $\mathcal{D}_{x,\epsilon}$ (see Appendix A.1).

## 2.2 CONSTRAINT SETS FROM F-DIVERGENCES

Dealing with the set $\mathcal{D}_{x,\epsilon}$ directly is difficult due to its possibly non-convex geometry. In this section, we discuss specific relaxations of this set, i.e., choices for sets $\mathcal{D}$ such that $\mathcal{D}_{x,\epsilon} \subseteq \mathcal{D}$ that are easier to optimize over. In particular, we focus on a general family of constraint sets defined in terms of $f$-divergences. These divergences satisfy a number of useful properties and include many well-known instances (e.g. relative entropy, total variation); see Appendix A.2 for details.

**Definition 2.3.** ($f$-divergence constraint set). Given $\rho, \nu \in \mathcal{P}(\mathcal{X})$, their $f$-divergence is defined as

$$D_f(\nu \| \rho) = \mathop{\mathrm{E}}_{X \sim \rho} \left[ f\left( \frac{\nu(X)}{\rho(X)} \right) \right] \ ,$$

where $f : \mathbb{R}_+ \mapsto \mathbb{R}$ is a convex function with $f(1) = 0$. Given a reference distribution $\rho$, an $f$-divergence $D_f$ and a bound $\epsilon_f \geq 0$, we define the *f-divergence constraint set* to be:

$$\mathcal{D}_f = \{\nu \in \mathcal{P}(\mathcal{X}) : D_f(\nu \| \rho) \leq \epsilon_f\} \ .$$

Technically, this definition depends on the Radon-Nikodym derivative of $\nu$ with respect to $\rho$, but we ignore measure-theoretic issues in this paper for simplicity of exposition. For continuous distributions, $\nu$ and $\rho$ should be treated as densities, and for discrete distributions as probability mass functions.

**Relaxations using $f$-divergence** This construction immediately allows us to obtain relaxations of $\mathcal{D}_{x,\epsilon}$. For example, by choosing $f(u) = u \log(u)$, we have the KL-divergence. Using KL-divergence yields the following relaxation between norm-based and divergence-based constraint sets for Gaussian smoothing measures, i.e. $\mu(x) = \mathcal{N}(x, \sigma^2 I)$:

$$\mathcal{D}_{x,\epsilon} = \{\mu(x') : \|x - x'\|_2 \leq \epsilon\} \subseteq \{\nu : \mathrm{KL}(\nu \| \mu(x)) \leq \epsilon^2/(2\sigma^2)\} \ .$$

Tighter relaxations can be constructed by combining multiple divergence-based constraints. In particular, suppose $\mathcal{F}$ is a collection of convex functions each defining an $f$-divergence, and assume each $f \in \mathcal{F}$ has a bound $\epsilon_f$ associated with it. Then we can define the constraint set containing perturbed distributions where all the bounds hold simultaneously (Fig. 1):

$$\mathcal{D}_{\mathcal{F}} := \bigcap_{f \in \mathcal{F}} \mathcal{D}_f = \{\nu : \forall f \in \mathcal{F} \ \ D_f(\nu \| \rho) \leq \epsilon_f\} \ .$$

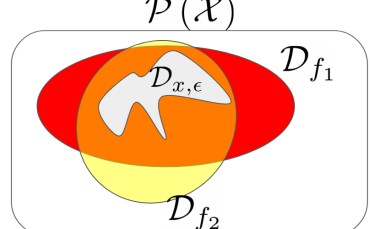

Figure 1: Intersecting $f$-divergence constraints to obtain better relaxations $\mathcal{D}_{\mathcal{F}}$ (depicted by the orange region) of $\mathcal{D}_{x,\epsilon}$.

---

[1] $d$ is an arbitrary distance function (not necessarily a metric e.g. ...)

In this paper, we work with the following divergences:

(1) Rényi: $R_\alpha(\nu\|\rho) = \log(1 + D_f(\nu\|\rho))/(\alpha - 1)$ where
$f(x) = x^\alpha - 1$ (for $\alpha \geq 1$), and $R_\alpha(\rho\|\nu) = \log(1 - D_f(\nu\|\rho))/(\alpha - 1)$ with $f(x) = 1 - x^\alpha$ (for $0 \leq \alpha \leq 1$). The limit $\alpha \to \infty$ yields the infinite order Rényi divergence $R_\infty(\nu\|\rho) = \sup_x(\nu(x)/\rho(x))$.

(2) KL: $\mathrm{KL}(\nu\|\rho) = D_f(\nu\|\rho)$ with $f(x) = x\log(x)$.

(3) Hockey-Stick: $D_{\mathrm{HS},\beta}(\nu\|\rho) = D_f(\nu\|\rho)$ with $f(x) = \max(x - \beta, 0) - \max(1 - \beta, 0)$.

It turns out that the Rényi and KL divergences are computationally attractive for a broad class of smoothing measures, while the Hockey-Stick divergences are theoretically attractive as they lead to optimal certificates in the information-limited setting. However, Hockey-Stick divergences are harder to estimate in general, so we only use them for Gaussian smoothing measures.

## 2.3 Computing f-divergence bounds

In general, our framework can be used with any family of smoothing measures and any family of $f$ divergences such that an upper bound on $\max_{\nu \in \mathcal{D}_{x,\epsilon}} D_f(\nu\|\rho)$ can be estimated efficiently. We describe how $f$-divergence bounds can be obtained for several classes of smoothing measures:

**Product measures**  Product measures are of the form $\mu(x) = \otimes_{i=1}^d \mu_i(x_i)$ where $\mathcal{X} = \prod_{i=1}^d \mathcal{X}_i$ and $\mu_i$ is a smoothing measure on $\mathcal{X}_i$. We note that the discrete smoothing measure used in (Lee et al., 2019), the Gaussian measure used in (Cohen et al., 2019) and the Laplacian measure used in (Li et al., 2019) are all of this form. For such measures, one can construct bounds on Rényi-divergences subject to any $\ell_p$ norm constraint using a Lagrangian relaxation of the optimization problem $\max_{x':\|x-x'\|_p \leq \epsilon} R_\alpha(\mu(x')\|\mu(x))$ (see Appendix A.3 for details).

**Norm-based smoothing measures**  Appendix A.9.1 also shows how we can obtain bounds on the infinite-order Rényi divergence $R_\infty$, as well as on several classes of $f$-divergences, for norm-based smoothing measures of the form $\mu(x)[X] \propto \exp(-\|X - x\|)$.

## 3 Our certification procedures

We now show how to reduce the problems of full-information and information-limited robust black-box certification to simple convex optimization problems for general constraint sets $\mathcal{D}$ defined in terms of $f$-divergences. This allows us, by extension, to solve the problem for related divergences like Rényi divergences. The following two theorems provide the main foundation for the verification procedures in the paper.

**Theorem 1** (Verifying full-information robust certification). Let $\mathcal{D}_\mathcal{F}$ be the constraint set defined by $\mathcal{F} = \{f_1, \ldots, f_M\}$ and $\epsilon_{f_i} = \epsilon_i$. Define $f_\lambda(u) = \sum_{i=1}^M \lambda_i f_i(u)$ and denote its convex conjugate[2] by $f_\lambda^*$. The specification $\phi$ is robustly certified at $\rho$ with respect to $\mathcal{D}_\mathcal{F}$ (cf. Definition 2.2) *if and only if* the optimal value of the following convex optimization problem is non-negative:

$$\max_{\lambda_1,\ldots,\lambda_M \geq 0, \kappa} \quad \kappa - \sum_{i=1}^M \lambda_i \epsilon_i - \mathop{\mathrm{E}}_{X \sim \rho}[f_\lambda^*(\kappa - \phi(X))] \ . \tag{6}$$

The proof of Theorem 1, given in Appendix A.4, uses standard duality results to show that the dual of the verification optimization problem has the desired form. We note that the special case where $M = 1$ reduces to Proposition 1 of Duchi & Namkoong (2018), although the result is used in a completely different context in that work.

To build a practical certification algorithm from Theorem 1, we must do two things: 1) compute the optimal values of $\lambda$ and $\kappa$; and 2) estimate the expectation in Eq. 6. Since the estimation of the expectation cannot be done in closed form (due to the black-box nature of $\phi$), we must rely on sampling. In step 1 of Algorithm 1, we use $N$ samples taken independently from $\rho$ to estimate the expectation and solve the "sampled" optimization problem using an off-the-shelf solver (Diamond

---

[2]For any function $f : \mathbb{R}_+ \mapsto \mathbb{R}$, its *convex conjugate* is defined as $f^*(u) = \max_{v \geq 0}(uv - f(v))$.

---

**Algorithm 1** Full information certification (see appendix A.9 for details of subroutines)

---

**Inputs:** Query access to specification $\phi : \mathcal{X} \to [a, b]$, sampling access to reference distribution $\rho$, divergences $f_i$ and bounds $\epsilon_i$, sample sizes $N, \tilde{N}$, confidence level $\zeta$.
1: $\kappa^*, \lambda^* \leftarrow \text{ESTIMATEOPT}(\rho, \phi, N, \{f_i\}_{i=1}^M, \{\epsilon_i\}_{i=1}^M)$.
2: $E_{ub} \leftarrow \text{UPPERCONFIDENCEBOUND}(\rho, \phi, \tilde{N}, \{f_i\}_{i=1}^M, \{\epsilon_i\}_{i=1}^M, a, b, \lambda^*, \kappa^*, \zeta)$.
3: If $\kappa^* - \sum_{i=1}^M \lambda_i^* \epsilon_i - E_{ub} \geq 0$ **return** CERTIFIED else **return** NOT CERTIFIED.

---

& Boyd, 2016). This gives us $\kappa^*, \lambda^*$, the estimated optimal values of $\kappa$ and $\lambda$, respectively. Then we take these values and compute a high-confidence lower bound on the objective function of Eq. 6, which is then used to verify robustness. In particular, in step 2, we compute a high-confidence upper bound $E_{ub}$ on the expectation term in the objective such that $E_{ub} \geq \mathrm{E}_{X \sim \rho}[f_\lambda^*(\kappa^* - \phi(X))]$ with probability at least $\zeta$; this computation involves taking $\tilde{N}$ independent samples from $\rho$ and finding a confidence interval around the resulting empirical estimate of the expectation (for details, see Eq. 25 in Appendix A.9.1). Plugging in this estimate back into Eq. 6 gives the desired high-confidence lower bound in step 3. Details of both subroutines ESTIMATEOPT and UPPERCONFIDENCEBOUND used in Algorithm 1 are given in Algorithm 3 in Appendix A.9.2.

Our next theorem concerns the specialization of this verification procedure to the information-limited setting.

**Theorem 2** (Verifying information-limited robust certification). Let $\mathcal{D}_\mathcal{F}$ be as in Theorem 1, and $S$ and $\theta_a, \theta_b$ be as in Definition 2.1. The class of specifications $S$ is information-limited robustly certified at $\rho$ with respect to $\mathcal{D}_\mathcal{F}$ (cf. Definition 2.1) *if and only if* the optimal value of the following convex optimization problem is non-negative:

$$\begin{aligned} \min_{\zeta_a, \zeta_b, \zeta_c \geq 0} \quad & \zeta_a - \zeta_b \\ \text{Subject to} \quad & \zeta_a + \zeta_b + \zeta_c = 1 \;, \quad D_{f_i}(\zeta \| \theta) \leq \epsilon_i \quad i = 1, \ldots, M \;, \end{aligned} \tag{7}$$

where $\theta = (\theta_a, \theta_b, 1 - \theta_a - \theta_b)$ and $\zeta = (\zeta_a, \zeta_b, \zeta_c)$ are interpreted as probability distributions.

The proof of Theorem 2 is presented in Appendix A.5. It is based on the fact that in the information-limited setting, it is possible to directly compute the expectation in Eq. 6, and in fact this expectation only depends on $\phi$ via the probabilities $\theta_a$ and $\theta_b$.

Theorem 2 naturally leads to a certification algorithm, presented in Algorithm 2. It simply uses the same procedure as Cohen et al. (2019, Section 3.2) to compute a high-confidence lower bound $\underline{\theta_a}$ on the probability of the correct class under randomized smoothing and then solves the convex optimization problem Eq. 7. Again, we can use an off-the-shelf solver CVXPY (Diamond & Boyd, 2016) in step 2 for the general $M > 1$ case, but closed-form solutions are also available for $M = 1$; these are given in Table 4 in Appendix A.6.

---

**Algorithm 2** Information-limited certification

---

**Inputs:** Query access to classifier $h$, correct label $y$, sampling access to reference distribution $\rho$, divergences $f_i$ and bounds $\epsilon_i$, sample sizes $N, \tilde{N}$, confidence level $\zeta$.
1: Sample $X^1, \ldots, X^N \sim \rho$ and test[3] whether $y = \arg\max_{y' \in \mathcal{Y}} \mathbb{P}_{X \sim \rho}[h(X) = y']$.
2: Sample $X^1, \ldots, X^{\tilde{N}} \sim \rho$ and compute[4] a bound $\mathbb{P}_{X \sim \rho}[h(X) = y'] \geq \underline{\theta_a}$ with confidence $\zeta$.
3: Obtain $o^*$ by solving Eq. 7 with $\theta_a \leftarrow \underline{\theta_a}$ and $\theta_b \leftarrow 1 - \underline{\theta_a}$.
4: If $o^* \geq 0$ **return** CERTIFIED else **return** NOT CERTIFIED.

---

## 4 THEORETICAL ANALYSIS OF CERTIFICATION METHODS

We now present theoretical results characterizing our certification methods and show the following:

---

[3]Using the algorithm from Hung & Fithian (2019).
[4]Using the algorithm from Clopper & Pearson (1934).

1. For smoothed probabilistic classifiers, the full-information certificate dominates the information-limited one.

2. In the information-limited setting, if we define the $f$-divergence relaxation $\mathcal{D}_{\mathcal{F}}$ using Hockey-Stick divergences with specific parameters, then the computed certificate is provably tight.

## 4.1 ADVANTAGE OF FULL-INFORMATION CERTIFICATION

Consider a soft binary classifier $H : \mathcal{X} \to [0, 1]$ that outputs the probability of label $+1$ and consider a point $x \in \mathcal{X}$ with $H(x) > 1/2$. We define the specification $\phi(x) = H(x) - \frac{1}{2}$. Then, the smoothed classifier $H_s(x) = \mathrm{E}_{X \sim \mu(x)}[H(X)]$ predicts label $+1$ for all $x'$ with $\|x' - x\| \leq \epsilon$ if and only if $\phi$ is full-information robustly certified at $\mu(x)$ with respect to $\mathcal{D}_{x,\epsilon} = \{\mu(x') : \|x' - x\| \leq \epsilon\}$. Note that the optimization in Theorem 1 depends on the full distribution of $\phi(X) \in [-1/2, 1/2]$, $X \sim \mu(x)$. On the other hand, to certify this robustness in the information-limited setting is equivalent to taking the specification $\phi(x) = 1[H(x) > 1/2]$ (the indicator function of the event $H(x) > 1/2$), in which case the only information available is $\theta_a = H_s(x) = \mathrm{E}_{X \sim \mu(x)}[H(X)]$.

To compare the two approaches, consider the objective of Eq. 6 with a single $f$-divergence constraint $D_f(\nu \| \rho) \leq \epsilon$. Then, we have

$$\kappa - \lambda\epsilon - \mathop{\mathrm{E}}_{X \sim \mu(x)}[f_\lambda^*(\kappa - \phi(X))] = \kappa - \lambda\epsilon - \mathop{\mathrm{E}}_{X \sim \mu(x)}\left[f_\lambda^*\left(\kappa - H(X) + \frac{1}{2}\right)\right]$$

$$= \kappa - \lambda\epsilon - \mathop{\mathrm{E}}_{X \sim \mu(x)}\left[f_\lambda^*\left(\left(\kappa + \frac{1}{2} - 1\right)H(X) + \left(\kappa + \frac{1}{2}\right)(1 - H(X))\right)\right]$$

$$\geq \kappa - \lambda\epsilon - \mathop{\mathrm{E}}_{X \sim \mu(X)}\left[f_\lambda^*\left(\kappa - \frac{1}{2}\right)H(X) + f_\lambda^*\left(\kappa + \frac{1}{2}\right)(1 - H(X))\right]$$

$$= \kappa - \lambda\epsilon - \theta_a f_\lambda^*\left(\kappa - \frac{1}{2}\right) - (1 - \theta_a)f_\lambda^*\left(\kappa + \frac{1}{2}\right) \ ,$$

where the third line follows from Jensen's inequality. The proof of Theorem 2 shows that maximizing the final expression above with respect to $\kappa, \lambda$ is equivalent to the dual of the information-limited certification problem Eq. 7. Thus, the information-limited setting computes a weaker certificate than the full-information setting for soft classifiers:

**Corollary 3.** The optimization problem of Eq. 6 with the specification $\phi$ defined above has an optimal value that is greater than or equal to that of the optimization problem defined in Eq. 7.

## 4.2 TIGHT RELAXATIONS FOR INFORMATION-LIMITED ROBUST CERTIFICATION

Ideally, we would like to certify robustness of specifications with respect to sets of the form $\mathcal{D}_{x,\epsilon} = \{\mu(x') : d(x, x') \leq \epsilon\}$. The following result shows that the gap between the ideal $\mathcal{D}_{x,\epsilon}$ and the tractable constraint sets $\mathcal{D}_{\mathcal{F}}$ can be closed *in the context of information-limited robust certification* provided that we can measure hockey-stick divergences of *every* non-negative order $\beta \geq 0$. The proof is given in Appendix A.7.

**Theorem 4.** Let $\theta_a, \theta_b, \rho, \mathcal{D}, S$ be as in Definition 2.1. Define $\epsilon_\beta = \max_{\nu \in \mathcal{D}} D_{\mathrm{HS},\beta}(\nu \| \rho)$ for all $\beta \geq 0$ and let $\beta_a^*, \beta_b^*$ be chosen as follows:

$$\beta_a^*, \beta_b^* = \mathop{\arg\max}_{\beta_a \geq \beta_b \geq 0} \ 1 - \beta_a(1 - \theta_a) - \beta_b\theta_b - \left(\epsilon_{\beta_a} + [1 - \beta_a]_+\right) - \left(\epsilon_{\beta_b} + [1 - \beta_b]_+\right) \ . \quad (8)$$

Define the constraint set

$$\mathcal{D}_{\mathrm{HS}} = \{\nu \in \mathcal{P}(\mathcal{X}) : D_{\mathrm{HS};\beta_a^*}(\nu \| \rho) \leq \epsilon_{\beta_a^*}\} \cap \{\nu \in \mathcal{P}(\mathcal{X}) : D_{\mathrm{HS};\beta_b^*}(\nu \| \rho) \leq \epsilon_{\beta_b^*}\} \ .$$

Then, $S$ is information-limited robustly certified at $\rho$ with respect to $\mathcal{D}$ *if and only if* $S$ is information-limited robustly certified at $\rho$ with respect to $\mathcal{D}_{\mathrm{HS}}$. Thus, the optimal information-limited certificate in this case can be obtained by applying theorem 2 to $\mathcal{D}_{\mathrm{HS}}$.

## 5 CONNECTIONS WITH PRIOR WORK

Table 1 summarizes the differences between our work and prior work in terms of the set of smoothing measures admitted, the offline computation cost of the certification procedure (which needs to

be performed once for every possible perturbation size and choice of smoothing measure), the perturbations considered, whether they can use information beyond $\theta_a, \theta_b$ to improve the certificates and whether they compute optimal certificates for a given smoothing measure in the information-limited setting.

| | $\mu(x)$ | Computation | $\|\cdot\|$ | Use additional information? | Optimal |
|---|---|---|---|---|---|
| Cohen et al. (2019) | Gaussian | $O(1)$ | $\ell_2$ | X | Yes |
| Lee et al. (2019) | Finite Support[*] | $O(d^3)$ | $\ell_0$ | Only decision trees | Yes |
| Li et al. (2019) | Gaussian Laplacian | $O(1)$ | $\ell_2$ $\ell_1$ | X | No |
| Lecuyer et al. (2019) | Arbitrary | $O(1)$ | All $\ell_p$ | X | No |
| Our work | Arbitrary | $O(1)$ | All $\ell_p$ | Arbitrary classifiers | Yes |

Table 1: Comparison between black-box verification methods. [*] Technically, Lee et al. (2019) can handle smoothing measures such that the likelihood ratios take a finite set of values, but most practical instances of this correspond to finite-support distributions.

**Cohen et al. (2019)** study the problem of verifying hard classifiers smoothed by Gaussian noise, and derive optimal certificates with respect to $\ell_2$ perturbations of the input. Their results can be recovered as a special case of our framework when applied to sets defined via constraints on hockey-stick divergences. Theorem 4 shows that the optimal certificate in the information-limited setting can be computed by applying theorem 2 to a constraint set with two hockey-stick divergences.

For the Gaussian measure $\mu(x) = \mathcal{N}(x, \sigma^2 I)$, the HS divergence $D_{\mathrm{HS}, \beta}(\mu(x) \| \mu(x'))$ can be computed in closed form and is purely a function of the $\ell_2$ distance $\|x - x'\|_2$. This enables us to efficienctly compute the $\beta_a^*, \beta_b^*$ in theorem 4. Thus, we obtain the following result (see Appendix A.7.2 for a proof):

**Corollary 5.** Let $\rho = \mathcal{N}(x, \sigma^2 I)$, $\mathcal{D}_{x,\epsilon} = \{\mathcal{N}(x', \sigma^2 I) : \|x - x'\|_2 \le \epsilon\}$ and $1 \ge \theta_a \ge \theta_b \ge 0$. Let $\mathcal{D}_{\mathrm{HS}}$ be defined as in theorem 4. Then, applying theorem 2 to the constraint set $\mathcal{D}_{\mathrm{HS}}$ gives the following condition for robust certification:

$$\Psi_g\left(\Psi_g^{-1}(\theta_a) - \frac{\epsilon}{\sigma}\right) + \Psi_g\left(\Psi_g^{-1}(1 - \theta_b) - \frac{\epsilon}{\sigma}\right) \ge 1 \ , \tag{9}$$

where $\Psi_g$ is the CDF of a standard normal random variable $\mathcal{N}(0, 1)$. With straightforward algebra (worked out in appendix A.7.2), this can be shown to be equivalent to

$$\Psi_g^{-1}(\theta_a) - \Psi_g^{-1}(\theta_b) \ge \frac{2\epsilon}{\sigma} \ ,$$

which is the certificate from Theorem 1 of Cohen et al. (2019).

**Lee et al. (2019)** derive optimal certificates in the information-limited setting under the assumption that the likelihood ratio between measures $\frac{\nu(X)}{\rho(X)}$ (where $\nu = \mu(x'), \rho = \mu(x)$) can only take values from a finite set. This is a restrictive assumption that prevents the authors from accommodating natural smoothing measures like Gaussian or Laplacian measures. Further, the complexity of computing the certificates in their framework is significant: $O(d^3)$ computation (where $d$ is the input dimension) is needed to certify smoothness to $\ell_0$ perturbations. The authors also derive tighter certificates for the special case of certain classes of decision trees by exploiting the tree structure. In contrast, our framework can derive tighter certificates in the full-information setting for arbitrary classifiers.

**Li et al. (2019)** use properties of Rényi divergences to derive robustness certificates for classifiers smoothed by Gaussian (resp. Laplacian) noise under $\ell_2$ (resp. $\ell_1$) perturbations. Their results can be obtained as special cases of ours; in particular, the Rényi divergence certificates in Table 4 (in Appendix A.6) recover the results of Lemma 1 of Li et al. (2019), but the latter are only applicable for Gaussian and Laplacian smoothing measures.

**Lecuyer et al. (2019)** introduce the notion of *pixel differential privacy* (pixelDP) and show that smoothing measures $\mu$ satisfying pixelDP with respect to a certain type of perturbations lead to adversarially robust classifiers. We can show that pixelDP can be viewed as a special instance of our

certification framework with two specific hockey-stick divergences, and that the certificates derived from the pixelDP are provably dominated by the certificates from our framework (Theorem 1) with the same choice of divergences (see Corollary 7 in Appendix A.7.3).

## 6 EXPERIMENTS

### 6.1 FULL-INFORMATION IMPROVES UPON INFORMATION-LIMITED CERTIFICATION

To compare *full-information certificates* with *limited-information certificates*, we trained a ResNet-152 model on ImageNet with data augmentation by adding noise via sampling from a zero-mean Gaussian with variance 0.5 for each coordinate; during certification we sample from the same distribution to estimate lower bounds on the probability of the top predicted class. For the full-information certificate, we use two hockey-stick divergences for the certificate and tune the parameters $\beta$ to obtain the highest value in the optimization problem in step 2 of Algorithm 1. For the infromation-limited certificate, our approach reduces to that of (Cohen et al., 2019) and we follow the same certification procedure. We use $N = 1000, \tilde{N} = 1000000, \zeta = .99$ for both certification procedures.

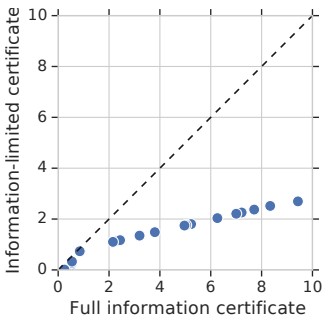

Figure 2: Information-limited vs full-information certificates for ImageNet for $\ell_2$ perturbations. The dashed line represents equal certificates and every point below the dashed line has a stronger certificate from the full information verification setting. We run the comparison on 50 randomly selected examples from the validation set. Each blue dot in Figure 2 corresponds to one test point, with its x coordinate representing the radius for full information certificate (from Algorithm 1) and y coordinate the information-limited certificate (which is equivalent to the certification procedure of Cohen et al., 2019). The running time of the full-information certification procedure is .2s per example (excluding the sampling cost) while the limited-information certification takes .002s per example. Both procedures incur the same sampling cost as they use the same number of samples.

Figure 2 shows the difference between the two certificates. The certificate provided by the full-information method is always stronger than the one given by the information-limited method. The difference is often substantial – for one of the test samples, the full-information setting can certify robustness to $\ell_2$ perturbations of radius $\epsilon = 9.42$ in the full-information case while the limited-information certificate can only be provided for perturbation radius $\epsilon = 2.69$.

### 6.2 SCALABILITY AND TIGHTNESS

In this section we consider $\ell_0$ perturbations for both ImageNet and Binary MNIST (that is, we consider the number of pixels that can be perturbed without changing the prediction). To test for scalability and tightness trade-offs of our framework, we compare our methodology to that of Lee et al. (2019), as their work obtains the optimal bound for $\ell_0$. We computed certificates for a single model for each classification task; for Binary MNIST we used the same model and training procedure as Lee et al. (2019) and for ImageNet, we used the model released in the Github code accompanying the paper of Lee et al. (2019). We use the discrete smoothing measure (appendix A.10) with parameter $p = 0.8$ for Binary MNIST certification, and $p = 0.2$ for ImageNet certification.

In our experiments we ran the certification procedures on all test examples from the Binary MNIST dataset, while for ImageNet, following prior work (Lee et al., 2019; Cohen et al., 2019), on every 100th example from validation set. The proportion of the examples for which $\epsilon$ accuracy can be certified are reported in Table 2 for various values of $\epsilon$. Comparing with the optimal certificates obtained by the certification method of Lee et al. (2019), the table shows that our bounds remains tight for Binary MNIST up until $\epsilon = 3$, and we do so with 130 times speed-up. For ImageNet, our bounds differ only about 7–15% in terms of accuracy, and we obtain this with approximately 20 million times speed-up.

| Dataset | Certificate | Smoothing Value | Computation Time | Certified Accuracy | | | | | | |
|---|---|---|---|---|---|---|---|---|---|---|
| | | | | $\epsilon = 1$ | $\epsilon = 2$ | $\epsilon = 3$ | $\epsilon = 4$ | $\epsilon = 5$ | $\epsilon = 6$ | $\epsilon = 7$ |
| Binary MNIST | Lee et al. (2019) | 0.8 | 3.68s | 0.919 | 0.767 | 0.530 | 0.513 | 0.348 | 0.194 | 0.095 |
| | Ours | | 0.028s | 0.919 | 0.767 | 0.530 | 0.296 | 0.162 | 0.080 | 0.015 |
| ImageNet | Lee et al. (2019) | 0.2 | 4 days | 0.488 | 0.418 | 0.310 | 0.250 | 0.244 | 0.234 | 0.224 |
| | Ours | | 0.028s | 0.362 | 0.262 | 0.224 | 0.186 | 0.136 | 0 | 0 |

Table 2: Proportion of the examples with verified $\ell_0$-robustness for different accuracy ($\epsilon$) parameters.

## 6.3 CERTIFICATION FOR AUDIO CLASSIFICATION: LIBRISPEECH

Audio classification systems have been shown to be susceptible to adversarial attacks (Qin et al., 2019). However, building audio classifiers that are provably robust to adversarial attacks has been hard due to the complexity of audio processing architectures. We take a step towards provably robust audio classifiers by showing that our approach can certify robustness of a classifier trained for speaker recognition on a state-of-the-art model for this task. We focus on $\ell_0$ perturbations that zero out a fraction of the audio sample, as they correspond to missing data in an audio signal. Missing data can occur due to errors in recording audio or packets dropped while transmitting an audio signal over a network and is a common issue (Turner, 2010; Smaragdis et al., 2009).

In principle, the method of Lee et al. (2019) is applicable to compute robustness certificates, but at an impractically large computational cost, since the computation needs to be repeated whenever an input of a new length (for which a certificate has not previously been computed) arrives. Concretely, this constitutes an $O(d^3)$ computation for the length $d$ ranging from 38 to 522,320 (the set of audio sequence lengths observed in the Librispeech test dataset (Panayotov et al., 2015)).

The results are shown in Table 3. To the best of our knowledge, these are the first results showing certified robustness of an audio classifier. We believe this is a significant advance towards certification of classifiers in audio and classifiers operating on variable-length inputs more generally.

| Dataset | Certificate | Smoothing Value | Computation Time | Certified Accuracy | | | | | | | |
|---|---|---|---|---|---|---|---|---|---|---|---|
| | | | | $\epsilon = 0$ | $\epsilon = 1$ | $\epsilon = 2$ | $\epsilon = 3$ | $\epsilon = 4$ | $\epsilon = 5$ | $\epsilon = 6$ | $\epsilon = 7$ |
| Librispeech | Ours | 0.5 | 0.028s | 0.511 | 0.225 | 0.091 | 0.091 | 0.091 | 0.091 | 0.091 | 0 |
| | | 0.7 | | 0.885 | 0.634 | 0.405 | 0.405 | 0.405 | 0.405 | 0 | 0 |
| | | 0.9 | | 0.872 | 0.772 | 0.711 | 0.711 | 0.711 | 0 | 0 | 0 |

Table 3: $\ell_0$ robustness results for Librispeech (Panayotov et al., 2015). From the Librispeech dataset, we created a corpus of sentence utterances from ten different speakers. The classification task is, given an audio sample, to predict whom is speaking. The test set consisted of 30 audio samples for each of the ten speakers. We use a DeepSpeaker architecture (Li et al., 2017), trained with the Adam optimizer ($\beta_1 = 0.9, \beta_2 = 0.5$) for 50,000 steps with a learning rate of 0.0001. The architecture is the same as that of Li et al. (2017), except for changing the number of neurons in the final layer for speaker identification with ten classes. Three models were trained with smoothing values of $p = 0.5$, $p = 0.7$, and $p = 0.9$, respectively, and we used the same values for certification. Certification was performed using $N = 1000, \tilde{N} = 1000000, \zeta = .99$ using $M = 1$ Rényi divergence, with $\alpha$ tuned to obtain the best certificate. The proportion of samples with certified robustness for different accuracy values are reported, computed on 300 test set samples.

## 7 CONCLUSION

We have introduced a general framework for black-box verification using $f$-divergence constraints. The framework improves upon state-of-the-art results on both image classification and audio tasks by a significant margin in terms of robustness certificates or computation time. We believe that our framework can potentially enable scalable computation of robustness verification for more complex predictors and structured perturbations that can be modeled using f-divergence constraints.

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

## A APPENDIX

### A.1 ADVERSARIAL SPECIFICATION FOR SMOOTHED CLASSIFIERS

Note that for any $\nu \in \mathcal{D}_{x,\epsilon}$ we have

$$\mathop{\mathrm{E}}_{X\sim\nu}[\phi_{c,c'}(X)] = \mathop{\mathbb{P}}_{X\sim\nu}[h(X) = c] - \mathop{\mathbb{P}}_{X\sim\nu}[h(X) = c'] \ .$$

Therefore, $\mathrm{E}_{X\sim\nu}[\phi_{c,c'}(X)] \geq 0$ for all $c' \in \mathcal{Y} \setminus \{c\}$ is equivalent to $c \in \arg\max_{y\in\mathcal{Y}} \mathbb{P}_{X\sim\nu}[h(X) = y]$. For $\nu = \mu(x')$, this means that $h_s(x') = c$ (assuming the argmax is unique). In other words, $\mathrm{E}_{X\sim\nu}[\phi_{c,c'}(X)] \geq 0$ for all $c' \in \mathcal{Y} \setminus \{c\}$ and all $\mu(x') \in \mathcal{D}_{x,\epsilon}$ if and only if $h_s(x') = c$ for all $x'$ such that $d(x, x') \leq \epsilon$, proving the required robustness certificate.

#### A.1.1 ROBUSTNESS SPECIFICATION FOR SMOOTHED SOFT CLASSIFIERS

Consider a soft classifier $H : \mathcal{X} \to \mathcal{P}(\mathcal{Y})$ that for each input $x$ returns a probability distribution $H(x)$ over the set of potential labels $\mathcal{Y}$ (e.g. $H$ might represent the outputs of the soft-max layer of a neural network). As in the case of hard classifiers, our methodology can be used to provide robustness guarantees for smoothed soft classifiers obtained by applying a smoothing measure $\mu(x)$ to the input. In this case, the smoothed classifier is again a soft classifier given by $H_s(x) = \mathrm{E}_{X\sim\mu(x)}[H(X)]$.

Let $x$ be a fixed input point and write $p = H_s(x) \in \mathcal{P}(\mathcal{Y})$ to denote the distribution over labels. A number of robustness properties about the soft classifier $H_s$ at $x$ can be phrased in terms of Definition 2.2. For example, let $\mathcal{Y} = \{1, \ldots, K\}$ and suppose that $p_1 \geq p_2 \geq \cdots \geq p_K$ so that $\{1, \ldots, k\}$ are the top $k$ labels at $x$. Then we can verify that the set of top $k$ labels will not change when moving the input from $x$ to $x'$ with $\|x - x'\| \leq \epsilon$ by defining the specifications $\phi_{i,j}(z) = H(z)_i - H(z)_j$ for $i \in [1, k]$ and $j \in [k + 1, K]$, and showing that all of these $\phi_{i,j}$ are robustly certified at $\mu(x)$ with respect to the set $\mathcal{D}_{x,\epsilon}$ defined above. The case $k = 1$ corresponds to robustness of the standard classification rule outputting the label with the largest score.

Another example is robustness of classifiers which are allowed to abstain. For example, suppose we build a hard classifier $\tilde{h}$ out of $H_s$ which returns the label with the maximum score as long as the gap between this score and the score of any other label is at least $\gamma$; otherwise it produces no output. Then we can certify that $\tilde{h}$ will not abstain and return the label $c = \arg\max_{y\in\mathcal{Y}} p_y$ at any point close to $x$ by showing that every $\phi_{c'}(z) = H(z)_c - H(z)_{c'} - \gamma$, $c' \neq c$, is robustly certified at $\mu(x)$ with respect to $\mathcal{D}_{x,\epsilon}$.

### A.2 BACKGROUND ON f-DIVERGENCES

A number of well-known properties about $f$-divergences are used throughout the paper, both explicitly and implicitly. Here we review such properties for the readers' convenience. Proofs and further details can be found in, e.g., (Csiszár et al., 2004; Liese & Vajda, 2006).

Recall that the $f$-divergences can be defined for any convex function $f : \mathbb{R}_+ \to \mathbb{R}$ such that $f(1) = 0$. We note that this requirement holds without loss of generality as the map $x \mapsto f(x) - f(1)$ is convex whenever $f$ is convex. Any $f$-divergence $D_f$ satisfies the following:

1. $D_f(\nu\|\rho) \geq 0$.
2. $D_f(\rho\|\rho) = 0$, and $D_f(\nu\|\rho) = 0$ implies $\nu = \rho$ whenever $f$ is strictly convex at 1.
3. $D_f(F_*(\nu)\|F_*(\rho)) \leq D_f(\nu\|\rho)$ for any function $F$, where $F_*(\rho)$ is the push-forward of $\rho$.
4. $D_f(\nu\|\rho) = D_{\bar{f}}(\rho\|\nu)$ where $\bar{f}(u) = uf\left(\frac{1}{u}\right)$ is again convex with $\bar{f}(1) = 0$.

### A.3 BOUNDING RENYI DIVERGENCES

We being with the optimization problem

$$\max_{x'} R_\alpha(\mu(x')\|\mu(x))$$
$$\text{subject to } \|x - x'\|_p \leq \epsilon \tag{10}$$

Since we have $R_\alpha(\mu(x')\|\mu(x)) = \sum_i R_\alpha(\mu_i(x_i')\|\mu_i(x_i))$. The constraint can be rewritten as

$$\sum_i |x_i - x_i'|^P \le \epsilon^p$$

Forming the Lagrangian relaxation, we obtain

$$\max_{x':|x_i'-x_i|\le\epsilon} \sum_i R_\alpha(\mu_i(x_i')\|\mu_i(x_i)) + \gamma\left(\sum_i |x_i' - x_i|^p - \epsilon^p\right) \ .$$

where the constraint $|x_i' - x_i| \le \epsilon$ is implied by $\|x' - x\|_p \le \epsilon$. We can maximize separately over each $x_i'$ to obtain

$$-\gamma\epsilon^p + \sum_i \max_{x_i'\in[x_i-\epsilon, x_i+\epsilon]} R_\alpha(\mu_i(x_i')\|\mu_i(x_i)) + \gamma|x_i' - x_i|^p \ .$$

By weak duality, for any $\gamma \ge 0$, this is an upper bound on Eq. 10. We can minimize this bound over $\gamma \ge 0$ to obtain the tightest bound.

The minimization over $x_i'$ for each $i$ can be solved in closed-form or via as simple 1-dimensional minimization problem for most smoothing measures.

### A.4 PROOF OF THEOREM 1

For simplicity of exposition (and to avoid measure theoretic issues), we focus on the case where $\nu, \rho$ have well defined densities $\nu(x), \rho(x)$ such that $\rho(x) > 0$ whenever $\nu(x) > 0$.

We begin by rewriting the optimization problem in terms of the likelihood ratio $r(X) = \frac{\nu(X)}{\rho(X)}$: We have

$$\mathop{\mathrm{E}}_{X\sim\nu}[\phi(X)] = \mathop{\mathrm{E}}_{X\sim\rho}[r(X)\phi(X)] \ , \quad D_{f_i}(\rho\|\nu) = \mathop{\mathrm{E}}_{X\sim\rho}[f_i(r(X))] \ , \quad \mathop{\mathrm{E}}_{X\sim\rho}[r(X)] = 1 \ ,$$

where the first two equalities follow directly by plugging in $\nu(X) = \rho(X)r(X)$ and the third is obtained using the fact that $\nu$ is a probability measure. Using these relations, the optimization over $\nu$ can be rewritten as

$$\min_{r\ge0} \mathop{\mathrm{E}}_{X\sim\rho}[r(X)\phi(X)]$$
$$\text{subject to} \quad \mathop{\mathrm{E}}_{X\sim\rho}[f_i(r(X))] \le \epsilon_i, \quad \mathop{\mathrm{E}}_{X\sim\rho}[r(X)] = 1 \ , \tag{11}$$

where $r \ge 0$ denotes that $r(x) \ge 0 \quad \forall x \in \mathcal{X}$. The optimization over $r$ is a convex optimization problem and can be solved using Lagrangian duality as follows – we first dualize the constraints on $r$ to obtain

$$\min_{r\ge0} \mathop{\mathrm{E}}_{X\sim\rho}[r(X)\phi(X)] + \sum_i \lambda_i\left(\mathop{\mathrm{E}}_{X\sim\rho}[f_i(r(X))] - \epsilon\right) + \kappa\left(1 - \mathop{\mathrm{E}}_{X\sim\rho}[r(X)]\right)$$

$$= \min_{r\ge0} \mathop{\mathrm{E}}_{X\sim\rho}\left[r(X)\phi(X) + \sum_i \lambda_i f_i(r(X)) - \kappa r(X)\right] + \kappa - \sum_i \lambda_i\epsilon$$

$$= \kappa - \sum_i \lambda_i\epsilon_i - \mathop{\mathrm{E}}_{X\sim\rho}\left[\max_{r\ge0} \kappa r - r\phi(X) - \sum_i \lambda_i f_i(r)\right]$$

$$= \kappa - \sum_i \lambda_i\epsilon_i - \mathop{\mathrm{E}}_{X\sim\rho}\left[\max_{r\ge0} \left(r(\kappa - \phi(X)) - f_\lambda(r)\right)\right]$$

$$= \kappa - \sum_i \lambda_i\epsilon_i - \mathop{\mathrm{E}}_{X\sim\rho}[f_\lambda^*(\kappa - \phi(X))] \ .$$

By strong duality, it holds that maximizing the final expression with respect to $\lambda \ge 0, \kappa$ achieves the optimal value in Eq. 11. Thus, if the optimal value is smaller than 0, the specification is not robustly certified and if it is larger than 0, the specification is robustly certified. Finally, since we are

ultimately interested in proving that the objective is non-negative, we can restrict ourselves to $\lambda \geq 0$ such that $\sum_i \lambda_i = 1$ (since if the optimal $\lambda$ added up to something larger, we could simply rescale the values to add up to 1 and multiply $\kappa$ by the same scaling factor without changing the sign of the objective function).

This concludes the proof of correctness of the certificate Eq. 6.

## A.5 PROOF OF THEOREM 2

For the next result, we observe that when $\phi$ is ternary valued, the optimization over $\kappa, \lambda$ above can be written as
$$\max_{\kappa, \lambda \geq 0} \kappa - \sum_i \lambda_i \epsilon_i - \theta_a f_\lambda^*(\kappa - 1) - \theta_b f_\lambda^*(\kappa + 1) - \theta_c f_\lambda^*(\kappa) \ ,$$
where $\theta_a = \mathbb{P}_{X \sim \rho}[\phi(X) = +1], \theta_b = \mathbb{P}_{X \sim \rho}[\phi(X) = -1], \theta_c = \mathbb{P}_{X \sim \rho}[\phi(X) = 0]$.

Writing out the expression for $f^*$, we obtain

$$\max_{\lambda \geq 0, \kappa} \min_{\gamma \geq 0} \kappa - \sum_i \lambda_i \epsilon_i - \theta_a \left( (\kappa - 1)\gamma_a - \sum_i \lambda_i f_i(\gamma_a) \right) - \theta_b \left( (\kappa + 1)\gamma_b - \sum_i \lambda_i f_i(\gamma_b) \right)$$
$$- \theta_c \left( \kappa \gamma_c - \sum_i \lambda_i f_i(\gamma_a) \right)$$

$$= \min_{\gamma \geq 0} \max_{\lambda \geq 0, \kappa} \kappa (1 - \theta_a \gamma_a - \theta_b \gamma_b - \theta_c \gamma_c) + \sum_i \lambda_i \left( \sum_{y \in \{a,b,c\}} \theta_y f_i(\gamma_y) - \epsilon_i \right) + \theta_a \gamma_a - \theta_b \gamma_b \ ,$$

where the second inequality follows from strong duality. The inner maximization is unbounded unless
$$\sum_{y \in \{a,b,c\}} \gamma_y \theta_y = 1 \ , \qquad \sum_{y \in \{a,b,c\}} \theta_y f_i(\gamma_y) \leq \epsilon_i \ .$$

One thing to note is that, we can rewrite these constraints in terms of $\zeta = \theta \odot \gamma$, i.e. $\zeta_y = \theta_y \gamma_y$ for $y \in \{a, b, c\}$. These constraints ensure that $\zeta$ is a probability distribution over $\{+1, 0, -1\}$ and furthermore
$$\sum_{y \in \{a,b,c\}} \theta_y f_i(\gamma_y) = D_{f_i}(\zeta \| \theta) \ .$$
Thus, the second constraint above is equivalent to $D_{f_i}(\zeta \| \theta) \leq \epsilon_i$. Writing the optimization problem in terms of $\zeta$, we obtain
$$\min_{\zeta_a, \zeta_b, \zeta_c \geq 0} \quad \zeta_a - \zeta_b$$
$$\text{subject to } D_{f_i}(\zeta \| \theta) \leq \epsilon_i \quad i = 1, \ldots, M \ ,$$
$$\zeta_a + \zeta_b + \zeta_c = 1 \ .$$

## A.6 CLOSED-FORM CERTIFICATES FOR THE INFORMATION-LIMITED SETTING

In this section we present closed-form certificates for the information-limited setting which can be derived from Theorem 2 for $M = 1$. The results are summarized in Table 4. In the next subsections we present the derivation of the certificates for Hockey-Stick and Rényi divergences. The certificates for the KL and infinite Rényi divergence can be derived by taking limits of the Rényi certificate (as $\alpha \to 1, \infty$ respectively).

### A.6.1 CALCULATION OF CERTIFICATE FOR HOCKEY-STICK DIVERGENCE

The function $f(u) = \max(u - \beta, 0) - \max(1 - \beta, 0)$ is a convex function with $f(1) = 0$. Then, we have
$$f_\lambda^*(u) = \max_{v \geq 0} (uv - \lambda \max(v - \beta, 0)) + \lambda \max(1 - \beta, 0)$$
$$= \begin{cases} \max(\beta u, 0) + \lambda \max(1 - \beta, 0) & \text{if } u \leq \lambda \ , \\ \infty & \text{if } u > \lambda \ . \end{cases}$$

| Divergence constraint | $f(u)$ | Certificate |
|---|---|---|
| KL divergence $\text{KL}(\nu\|\rho) \leq \epsilon_{KL}$ | $u\log(u)$ | $\epsilon_{KL} \leq -\log\left(1 - \left(\sqrt{\theta_a} - \sqrt{\theta_b}\right)^2\right)$ |
| Rényi divergences ($\alpha \geq 0$) $R_\alpha(\nu\|\rho) \leq \epsilon_{R,\alpha}$ | $\text{sign}(\alpha-1)(u^\alpha - 1)$ | $\epsilon_{R,\alpha} \leq -\log(1 - \theta_a - \theta_b + 2\eta)$ $\eta = \left(\frac{\theta_a{}^{(1-\alpha)} + \theta_b{}^{(1-\alpha)}}{2}\right)^{\left(\frac{1}{1-\alpha}\right)}$ |
| Infinite Rényi divergence $R_\infty(\nu\|\rho) \leq \epsilon_{R,\infty}$ | – | $\epsilon_{R,\infty} \leq -\log(1 - (\theta_a - \theta_b))$ |
| Hockey-stick divergences ($\beta \geq 0$) $D_{\text{HS},\beta}(\nu\|\rho) \leq \epsilon_{\text{HS},\beta}$ | $[u - \beta]_+ - [1 - \beta]_+$ | $\epsilon_{\text{HS},\beta} \leq \left[\frac{\beta(\theta_a - \theta_b) - |\beta - 1|}{2}\right]_+$ |

Table 4: Certificates for various $f$-divergences for the information-limited setting. Note that the Rényi divergences are not proper $f$-divergences, but are defined as $R_\alpha(\nu\|\rho) = \frac{1}{\alpha-1}\log(1 + D_f(\nu\|\rho))$. The infinite Rényi divergence, defined as $\sup_x \log(\nu(x)/\rho(x))$, is obtained by taking the limit $\alpha \to \infty$. All certificates depend on the gap between $\theta_a$ and $\theta_b$. Notation: $[u]_+ = \max(u, 0)$.

The certificate given by Eq. 6 in Theorem 1 for this divergence in the case of a smoothed hard classifier takes the form

$$\max_{\kappa \in \mathbb{R}, \lambda \geq 0}\left(\kappa - \mathop{\mathrm{E}}_{X \sim \rho}\left[f_\lambda^*(\kappa - \phi(X))\right]\right) - \lambda\epsilon \geq 0 \ ,$$

where the specification takes the values

$$\phi(X) = \begin{cases} +1 & \text{w.p. } \theta_a \ , \\ -1 & \text{w.p. } \theta_b \ , \\ 0 & \text{w.p. } 1 - \theta_a - \theta_b \ . \end{cases}$$

Plugging in the expression for $f^*$ the objective function above takes the form

$$\kappa - \beta\left(\theta_a[\kappa - 1]_+ + \theta_b[\kappa + 1]_+ + (1 - \theta_a - \theta_b)[\kappa]_+\right) - \lambda(\epsilon + \max(1 - \beta, 0)) \ ,$$

where we use the notation $[u]_+ = \max(u, 0)$ and assumed the constraints $\kappa \leq \lambda - 1$ since the objective is $-\infty$ otherwise. If $\beta \leq 1$, the objective is increasing monotonically in $\kappa$, so the optimal value is to set $\kappa$ to its upper bound $\lambda - 1$. Plugging this in, the possible values of the derivative with respect to $\lambda$ are

$$\begin{cases} \beta(1 - \theta_b) - \epsilon & \text{if } 0 \leq \lambda < 1 \ , \\ \beta\theta_a - \epsilon & \text{if } 1 < \lambda < 2 \ , \\ -\epsilon & \text{if } \lambda > 2 \ . \end{cases}$$

Thus, if $\epsilon \leq \beta\theta_a$, the maximum is attained at 2, if $\beta\theta_a \leq \epsilon \leq \beta(1 - \theta_b)$, the maximum is attained at 1, else the maximum is attained at 0, leading to the certificate:

$$\begin{cases} -1 & \text{if } \epsilon \geq \beta(1 - \theta_b) \ , \\ \beta(1 - \theta_b) - \epsilon - 1 & \text{if } \beta\theta_a \leq \epsilon \leq \beta(1 - \theta_b) \ , \\ \beta(1 + (\theta_a - \theta_b)) - 2\epsilon - 1 & \text{if } \epsilon \leq \beta\theta_a \ . \end{cases}$$

Thus, the certificate is non-negative only if

$$\epsilon \leq \max\left(\frac{\beta(1 + (\theta_a - \theta_b)) - 1}{2}, 0\right) \ .$$

The case $\beta \geq 1$ can be worked out similarly, leading to

$$\epsilon \leq \max\left(\frac{\beta(-1 + (\theta_a - \theta_b)) + 1}{2}, 0\right) \ .$$

The two cases can be combined as

$$\epsilon \leq \max\left(\frac{\beta(\theta_a - \theta_b) - |\beta - 1|}{2}, 0\right) \ .$$

### A.6.2 Calculation of certificate for Rényi divergence

We consider the cases $\alpha \geq 1$ and $\alpha \leq 1$ separately.

**Case 1** $(\alpha \geq 1)$    If $\alpha \geq 1$, the function $f(u) = (u^\alpha - 1)$ is a convex function with $f(1) = 0$. Then, we have

$$f_\lambda^*(u) = \max_{v \geq 0} uv - \lambda(v^\alpha - 1) = \begin{cases} \lambda & \text{if } u \leq 0 \\ \lambda + \lambda(\alpha - 1)\left(\frac{u}{\lambda\alpha}\right)^{\frac{\alpha}{\alpha-1}} & \text{if } u \geq 0 \end{cases}$$

$$= \lambda + \lambda(\alpha - 1)\left(\frac{\max(u, 0)}{\lambda\alpha}\right)^{\frac{\alpha}{\alpha-1}} .$$

Suppose we have a bound on the Rényi divergence $R_\alpha(\nu\|\rho) \leq \epsilon$. Then we know $D_f(\nu\|\rho) \leq \exp((\alpha - 1)\epsilon) - 1$. Let $\beta = \frac{\alpha}{\alpha-1}$ and

$$B = \theta_a(\max(0, \kappa - 1))^\beta + \theta_b(\max(0, \kappa + 1))^\beta + (1 - \theta_a - \theta_b)(\max(0, \kappa))^\beta .$$

Then the certificate Eq. 6 simplifies to (after some algebra)

$$\max_{\lambda \geq 0, \kappa} \kappa - \lambda \exp((\alpha - 1)\epsilon) - B\lambda^{1-\beta}\frac{(\alpha - 1)}{\alpha^\beta} .$$

Setting the derivative with respect to $\lambda$ to 0 and solving for $\lambda$, we obtain

$$\lambda = \frac{1}{\alpha}\left(\frac{B}{\exp((\alpha - 1)\epsilon)}\right)^{\left(\frac{1}{\beta}\right)} ,$$

and the optimal certificate reduces to

$$\max_\kappa \kappa - B^{\frac{1}{\beta}} \exp\left(\frac{\epsilon}{\beta}\right) .$$

For this number to be positive, we need that $\kappa \geq 0$ and

$$\frac{\kappa}{B^{\frac{1}{\beta}}} \geq \exp\left(\frac{\epsilon}{\beta}\right) .$$

The LHS above evaluates to

$$\left(\theta_a \max(0, 1 - \gamma)^\beta + \theta_b \max(0, 1 + \gamma)^\beta + 1 - \theta_a - \theta_b\right)^{-\frac{1}{\beta}} .$$

where $\gamma = \frac{1}{\kappa} \geq 0$. Maximizing this expression with respect to $\gamma$, we obtain

$$\gamma = \frac{\theta_a^{\alpha-1} - \theta_b^{\alpha-1}}{\theta_a^{\alpha-1} + \theta_b^{\alpha-1}} ,$$

so that the certificate reduces to

$$\left(2^\beta \theta_a \theta_b \left(\theta_a^{\alpha-1} + \theta_b^{\alpha-1}\right)^{\left(-\frac{1}{\alpha-1}\right)} + 1 - \theta_a - \theta_b\right)^{\left(\frac{1}{\beta}\right)} \geq \exp\left(\frac{\epsilon}{\beta}\right) .$$

Taking logarithms now gives the result.

**Case 2** $(0 \leq \alpha \leq 1)$    When $0 \leq \alpha \leq 1$, the function $f(u) = (1 - u^\alpha)$ is a convex function with $f(1) = 0$. Then, we have

$$f_\lambda^*(u) = \max_{v \geq 0} uv - \lambda(1 - v^\alpha) = \begin{cases} -\lambda + \lambda^{\frac{1}{1-\alpha}}(1 - \alpha)\left(\frac{-u}{\alpha}\right)^{\left(-\frac{\alpha}{1-\alpha}\right)} & \text{if } u \leq 0 , \\ \infty & \text{otherwise} . \end{cases}$$

Further, a bound $R_\alpha(\nu\|\rho) \leq \epsilon$ implies

$$D_f(\nu\|\rho) \leq 1 - \exp((\alpha - 1)\epsilon) .$$

Then the certificate from Eq. 6 reduces to

$$\max_{\kappa, \lambda \geq 0} \kappa + \lambda \exp((\alpha - 1)\epsilon) - (1 - \alpha)\lambda^{\frac{1}{1-\alpha}} \alpha^{\frac{\alpha}{1-\alpha}} \left( \theta_a (1 - \kappa)^{-\frac{\alpha}{1-\alpha}} + \theta_b(-1 - \kappa)^{-\frac{\alpha}{1-\alpha}} + \theta_c(-\kappa)^{-\frac{\alpha}{1-\alpha}} \right)$$

with the constraint $\kappa \leq -1$ (otherwise the certificate is $-\infty$). Setting the derivative with respect to $\lambda$ to 0 and solving for $\lambda$, we obtain

$$\lambda = \frac{\exp\left((\alpha - 1)\epsilon\left(\frac{1-\alpha}{\alpha}\right)\right)}{\alpha\omega} \quad ,$$

where

$$\omega = \left( \theta_a (1 - \kappa)^{-\frac{\alpha}{1-\alpha}} + \theta_b(-1 - \kappa)^{-\frac{\alpha}{1-\alpha}} + \theta_c(-\kappa)^{-\frac{\alpha}{1-\alpha}} \right)^{\left(\frac{1-\alpha}{\alpha}\right)} \quad .$$

Plugging this back into the certificate and setting $\beta = \frac{\alpha}{1-\alpha}$, we obtain

$$\kappa + \frac{\exp\left(-\frac{\epsilon}{\beta}\right)}{\omega} \quad .$$

For this number to be positive, we require that

$$\frac{1}{-\kappa\omega} \geq \exp\left(\frac{\epsilon}{\beta}\right) \quad .$$

The LHS of the above expression evaluates to

$$\left( \theta_a(1 + \gamma)^{(-\beta)} + \theta_b(1 - \gamma)^{(-\beta)} + 1 - \theta_a - \theta_b \right)^{\left(\frac{-1}{\beta}\right)} \quad ,$$

where $\gamma = -\frac{1}{\kappa}$. Maximizing this expression over $\gamma \in [0, 1]$, we obtain the final certificate to be

$$\left( 1 - \theta_a - \theta_b + 2\left( \frac{\theta_a^{1-\alpha} + \theta_b^{1-\alpha}}{2} \right)^{\left(\frac{1}{1-\alpha}\right)} \right)^{\left(-\frac{1}{\beta}\right)} \geq \exp\left(\frac{\epsilon}{\beta}\right) \quad .$$

Taking logarithms, we obtain

$$\epsilon \leq -\log\left( 1 - \theta_a - \theta_b + 2\left( \frac{\theta_a^{1-\alpha} + \theta_b^{1-\alpha}}{2} \right)^{\left(\frac{1}{1-\alpha}\right)} \right) \quad .$$

## A.7 Information-limited robust certification and tight relaxations

### A.7.1 Proof of Theorem 4

At a high level, the proof shows that, in the information-limited case, to achieve robust certification under an arbitrary set of constraints $\mathcal{D}$ it suffices to know the "envelope" of $\mathcal{D}$ with respect to all hockey-stick divergences of order $\beta \geq 0$, i.e. the function $\beta \mapsto \max_{\nu \in \mathcal{D}} D_{\text{HS}, \beta}(\nu \| \rho)$ captures all the necessary information to provide information-limited robust certification with respect to $\mathcal{D}$.

We start by considering the following optimization problem:

$$\min_{\Psi: \mathcal{X} \mapsto \{-1, 0, +1\}, \nu \in \mathcal{D}} \mathop{\mathrm{E}}_{X \sim \nu}[\Psi(X)]$$
$$\text{subject to} \quad \mathop{\mathrm{E}}_{X \sim \rho}[\mathbb{1}[\Psi(X) = +1]] \geq \theta_a \quad , \tag{12}$$
$$\mathop{\mathrm{E}}_{X \sim \rho}[\mathbb{1}[\Psi(X) = -1]] \leq \theta_b \quad .$$

In the information-limited setting, this problem attains the minimum expected value over $\phi \in S$. Here $\mathbb{1}[\phi(X) = 1]$ denotes the indicator function.

It will be convenient to write this in a slightly different form: Rather than looking at the outputs of $\Psi$ as the $+1, 0, -1$, we look at them as vectors in $\mathbb{R}^3$:

$$\mathcal{Z} = \left\{ \begin{pmatrix} 1 \\ 0 \\ 0 \end{pmatrix}, \begin{pmatrix} 0 \\ 1 \\ 0 \end{pmatrix}, \begin{pmatrix} 0 \\ 0 \\ 1 \end{pmatrix} \right\}$$

and define

$$\mathbf{a} = \begin{pmatrix} 1 \\ 0 \\ -1 \end{pmatrix}, \quad \mathbf{a}_+ = \begin{pmatrix} 1 \\ 0 \\ 0 \end{pmatrix}, \quad \mathbf{a}_- = \begin{pmatrix} 0 \\ 0 \\ 1 \end{pmatrix}.$$

Then, we can write the optimization problem Eq. 12 equivalently as

$$\min_{\Psi:\mathcal{X}\mapsto\mathcal{Z}, \nu\in\mathcal{D}} \operatorname*{E}_{X\sim\nu} \left[ \mathbf{a}^T \Psi(X) \right]$$
$$\text{subject to } \operatorname*{E}_{X\sim\rho} \left[ \mathbf{a}_+{}^T \Psi(X) \right] \geq \theta_a \ , \tag{13}$$
$$\operatorname*{E}_{X\sim\rho} \left[ \mathbf{a}_-{}^T \Psi(X) \right] \leq \theta_b \ .$$

We first consider the minimization over $\Psi$ for a fixed value of $\nu$. We begin by observing that since the objective is linear, the optimization over $\Psi$ can be replaced with the optimization over the convex hull of the set of $\Psi$ that satisfy the constraints (Bubeck, 2013). Since each input $x \in \mathcal{X}$ can be mapped independently of the rest, the convex hull is simply the cross product of the convex hull at every $x$, to obtain the constraint set

$$\left\{ \Psi : \mathcal{X} \mapsto \mathcal{P}(\mathcal{Z}) \text{ such that } \operatorname*{E}_{X\sim\rho} \left[ \mathbf{a}_+^T \Psi(X) \right] \geq \theta_a, \operatorname*{E}_{X\sim\rho} \left[ \mathbf{a}_-^T \Psi(X) \right] \leq \theta_b \right\} \ .$$

Therefore, the optimization problem reduces to

$$\min_{\Psi:\mathcal{X}\mapsto\mathcal{P}(\mathcal{Z})} \operatorname*{E}_{X\sim\nu} \left[ \mathbf{a}^T \Psi(X) \right]$$
$$\text{subject to } \operatorname*{E}_{X\sim\rho} \left[ \mathbf{a}_+{}^T \Psi(X) \right] \geq \theta_a \ , \tag{14}$$
$$\operatorname*{E}_{X\sim\rho} \left[ \mathbf{a}_-{}^T \Psi(X) \right] \leq \theta_b \ .$$

This is a convex optimization problem in $\Psi$. Denote

$$r(X) = \frac{\nu(X)}{\rho(X)} \ .$$

Considering the dual of this optimization problem with respect to the optimization variable $\Psi$, we obtain

$$\min_{\Psi} \operatorname*{E}_{X\sim\rho} \left[ \mathbf{a}^T \Psi(X) r(X) \right] - \lambda_a \left( \operatorname*{E}_{X\sim\rho} \left[ \mathbf{a}_+{}^T \Psi(X) \right] - \theta_a \right) + \lambda_b \left( \operatorname*{E}_{X\sim\rho} \left[ \mathbf{a}_-{}^T \Psi(X) \right] - \theta_b \right)$$
$$= \min_{\Psi} \lambda_a \theta_a - \lambda_b \theta_b + \operatorname*{E}_{X\sim\rho} \left[ (r(X)\mathbf{a} - \lambda_a \mathbf{a}_+ + \lambda_b \mathbf{a}_-)^\top \Psi(X) \right]$$
$$= \min_{\Psi} \lambda_a \theta_a - \lambda_b \theta_b + \operatorname*{E}_{X\sim\rho} \left[ \begin{pmatrix} r(X) - \lambda_a \\ 0 \\ -r(X) + \lambda_b \end{pmatrix}^T \Psi(X) \right] \ .$$

Since we can choose $\Psi(x)$ independently for each $x \in \mathcal{X}$, we can minimize each term in the expectation independently to obtain

$$\min_{\Psi(x)\in\mathcal{P}(\mathcal{Z})} \begin{pmatrix} r(x) - \lambda_a \\ 0 \\ r(x) + \lambda_b \end{pmatrix}^T \Psi(x) = \min(r(x) - \lambda_a, 0, -r(x) + \lambda_b) \ .$$

This implies that the Lagrangian evaluates to

$$\lambda_a \theta_a - \lambda_b \theta_b + \operatorname*{E}_{X\sim\rho} \left[ \min(r(X) - \lambda_a, 0, r(X) + \lambda_b) \right] \ .$$

We now consider two cases:

**Case 1** ($\lambda_a \geq \lambda_b \geq 0$)  In this case, we can see that

$$\min(r(X) - \lambda_a, 0, -r(X) + \lambda_b) = \min(r(X) - \lambda_a, 0) + \min(-r(X) + \lambda_b, 0)$$
$$= r(X) - \lambda_a - \max(r(X) - \lambda_a, 0) - \max(r(X) - \lambda_b, 0) \ .$$

Then, the Lagrangian reduces to

$$\lambda_a \theta_a - \lambda_b \theta_b + \operatorname*{E}_{X \sim \rho}[r(X) - \lambda_a] - \operatorname*{E}_{X \sim \rho}[\max(r(X) - \lambda_a, 0)] - \operatorname*{E}_{X \sim \rho}[\max(r(X) - \lambda_b, 0)]$$
$$= 1 - \lambda_a(1 - \theta_a) - \lambda_b \theta_b - (D_{\mathrm{HS},\lambda_a}(\nu\|\rho) + \max(1 - \lambda_a, 0)) - (D_{\mathrm{HS},\lambda_b}(\nu\|\rho) + \max(1 - \lambda_b, 0)) \ .$$

**Case 2** ($\lambda_b \geq \lambda_a \geq 0$)  In this case, we can see that

$$\min(r(X) - \lambda_a, 0, -r(X) + \lambda_b) = \min(r(X) - \lambda_a, -r(X) + \lambda_b)$$
$$= r(X) - \lambda_a + 2\min\left(0, \frac{\lambda_a + \lambda_b}{2} - r(X)\right)$$
$$= r(X) - \lambda_a - 2\max\left(r(X) - \frac{\lambda_a + \lambda_b}{2}, 0\right) \ .$$

Then, the Lagrangian reduces to

$$\lambda_a \theta_a - \lambda_b \theta_b + \operatorname*{E}_{X \sim \rho}[r(X) - \lambda_a] - 2\operatorname*{E}_{X \sim \rho}\left[\max\left(r(X) - \frac{\lambda_a + \lambda_b}{2}, 0\right)\right]$$
$$= 1 - \lambda_a(1 - \theta_a) - \lambda_b \theta_b - 2\left(D_{\mathrm{HS},\frac{\lambda_a + \lambda_b}{2}}(\nu\|\rho) + \max\left(1 - \frac{\lambda_a + \lambda_b}{2}, 0\right)\right) \ .$$

We know that $1 - \theta_a \geq \theta_b$ and $\lambda_b \geq \lambda_a$. If $\lambda_b > \lambda_a$, by choosing $\lambda_a' = \lambda_a + \kappa$ and $\lambda_b' = \lambda_b - \kappa$ for some small $\kappa > 0$, we know that the the sum of the first three terms would reduce while the final term would remain unchanged. Thus, at the the optimum in this case, we can assume $\lambda_a = \lambda_b$ and we obtain

$$1 - \lambda_a(1 - \theta_a) - \lambda_a \theta_b - 2(D_{\mathrm{HS},\lambda_a}(\nu\|\rho) + \max(1 - \lambda_a, 0)) \ .$$

**Final analysis of the Lagrangian**  Combining the two cases we can write the dual problem as

$$\max_{\lambda_a \geq \lambda_b \geq 0} 1 - \lambda_a(1 - \theta_a) - \lambda_b \theta_b - (D_{\mathrm{HS},\lambda_a}(\nu\|\rho) + \max(1 - \lambda_a, 0))$$
$$- (D_{\mathrm{HS},\lambda_b}(\nu\|\rho) + \max(1 - \lambda_b, 0)) \ . \tag{15}$$

By strong duality, the optimal value of the above problem precisely matches the optimal value of Eq. 14 (and hence Eq. 12). Thus, information limited robust certification with respect to $\mathcal{D}$ holds if and only if Eq. 15 has a non-negative optimal value for each $\nu \in \mathcal{D}$. Since we have that

$$\max_{\nu \in \mathcal{D}} D_{\mathrm{HS},\lambda_a}(\nu\|\rho) = \epsilon_{\lambda_a} \ , \quad \max_{\nu \in \mathcal{D}} D_{\mathrm{HS},\lambda_b}(\nu\|\rho) = \epsilon_{\lambda_b} \ ,$$

information-limited robust certification holds if and only if the optimal value of

$$\max_{\lambda_a \geq \lambda_b \geq 0} 1 - \lambda_a(1 - \theta_a) - \lambda_b \theta_b - (\epsilon_{\lambda_a} + \max(1 - \lambda_a, 0)) - (\epsilon_{\lambda_b} + \max(1 - \lambda_b, 0)) \tag{16}$$

is non-negative. Further, since the optimal value only depended on the value of $D_{\mathrm{HS},\beta}(\nu\|\rho)$ for $\beta \geq 0$, it is equivalent to information-limited robust certification with respect to $\mathcal{D}_{\mathrm{HS}}$.

The above argument also shows that in this case, information-limited robust certification with respect to $\mathcal{D}$ is equivalent to requiring that the following convex optimization problem has a non-negative optimal value:

$$\max_{\lambda_a \geq \lambda_b \geq 0} 1 - \lambda_a(1 - \theta_a) - \lambda_b \theta_b - \left(\epsilon_{\lambda_a} + [1 - \lambda_a]_+\right) - \left(\epsilon_{\lambda_b} + [1 - \lambda_b]_+\right) \ . \tag{17}$$

Let $\lambda_a^*, \lambda_b^*$ be the optimal values attained. Since this certificate depends only on the value of two Hockey-stick divergences at $\lambda_a^*, \lambda_b^*$, it must coincide with the application of theorem 2 to the constraint set $\mathcal{D}_{\mathrm{HS}}$ defined by constraints on these hockey-stick divergences (as we know that 2 computes the optimal certificate for any constraint set defined only by a set of f-divergences). This observation completes the proof.

### A.7.2 GAUSSIAN SMOOTHING MEASURES

Theorem 4 gives us the optimal limited-information certificate problem provided that we can compute

$$\max_{x':d(x,x')\leq \epsilon} D_{\mathrm{HS},\beta}(\mu(x')\|\mu(x))$$

for each $\beta \geq 0$. In particular, when $\mu$ is a Gaussian measure $\mu(x) = \mathcal{N}(x, \sigma^2 I)$, we can leverage the following result from Balle & Wang (2018).

**Lemma 6.** Let $\Psi_g$ be the CDF of a standard normal random variable $\mathcal{N}(0, 1)$. For any $\beta \geq 0$ and $x \in \mathbb{R}^d$ we have

$$\max_{x':\|x-x'\|_2\leq \epsilon} D_{\mathrm{HS},\beta}(\mu(x')\|\mu(x)) = \Psi_g\left(\frac{\epsilon}{2\sigma} - \frac{\log(\beta)\sigma}{2\epsilon}\right) - \beta\Psi_g\left(-\frac{\epsilon}{2\sigma} - \frac{\log(\beta)\sigma}{2\epsilon}\right) - [1-\beta]_+ \ .$$

Applying Eq. 17 to the expression in Lemma 6 proves Corollary 5.

*Proof of Corollary 5.* With the notation from Theorem 4 we have, for $\beta \geq 0$,

$$\epsilon_\beta = \Psi_g\left(\frac{\epsilon}{2\sigma} - \frac{\log(\beta)\sigma}{2\epsilon}\right) - \beta\Psi_g\left(-\frac{\epsilon}{2\sigma} - \frac{\log(\beta)\sigma}{2\epsilon}\right) - \max(1-\beta, 0) \ .$$

Plugging this expression into Eq. 17 allows us to verify information-limited robust certification of $\mathcal{N}(x', \sigma^2 I)$ with respect to $\mathcal{D}_{x,\epsilon} = \{\mathcal{N}(x', \sigma^2 I) : \|x - x'\|_2 \leq \epsilon\}$ by solving

$$\max_{\lambda_a\geq\lambda_b\geq 0} 1 - \lambda_a(1-\theta_a) - \lambda_b\theta_b$$
$$- \left(\Psi_g\left(\frac{\epsilon}{2\sigma} - \frac{\log(\lambda_a)\sigma}{2\epsilon}\right) - \lambda_a\Psi_g\left(-\frac{\epsilon}{2\sigma} - \frac{\log(\lambda_a)\sigma}{2\epsilon}\right)\right)$$
$$- \left(\Psi_g\left(\frac{\epsilon}{2\sigma} - \frac{\log(\lambda_b)\sigma}{2\epsilon}\right) - \lambda_b\Psi_g\left(-\frac{\epsilon}{2\sigma} - \frac{\log(\lambda_b)\sigma}{2\epsilon}\right)\right) \ .$$

Eq. 9 then follows from setting the derivatives of this expression to 0 with respect to $\lambda_a, \lambda_b$ and imposing the condition that the optimal solution is non-negative. $\square$

To check that Corollary 5 is equivalent to the optimal certification in (Cohen et al., 2019, Theorem 1) we first recall that, in our notation, their result can be stated as: the class of specifications $S$ in Definition 2.1 is information-limited robustly certified at $\rho = \mathcal{N}(x, \sigma^2 I)$ with respect to $\mathcal{D}_{x,\epsilon} = \{\mathcal{N}(x', \sigma^2 I) : \|x - x'\|_2 \leq \epsilon\}$ if and only if

$$\frac{2\epsilon}{\sigma} \leq \Psi_g^{-1}(\theta_a) - \Psi_g^{-1}(\theta_b) \ . \tag{18}$$

The equivalence between Eq. 9 and Eq. 18 now follows from the identity $1 - \Psi_g(\theta) = \Psi_g(-\theta)$ and the monotonicity of $\Psi_g$:

$$\Psi_g\left(\Psi_g^{-1}(\theta_a) - \frac{\epsilon}{\sigma}\right) + \Psi_g\left(\Psi_g^{-1}(1-\theta_b) - \frac{\epsilon}{\sigma}\right) \geq 1$$
$$\iff \Psi_g\left(\Psi_g^{-1}(\theta_a) - \frac{\epsilon}{\sigma}\right) \geq 1 - \Psi_g\left(\Psi_g^{-1}(1-\theta_b) - \frac{\epsilon}{\sigma}\right)$$
$$\iff \Psi_g\left(\Psi_g^{-1}(\theta_a) - \frac{\epsilon}{\sigma}\right) \geq \Psi_g\left(\frac{\epsilon}{\sigma} - \Psi_g^{-1}(1-\theta_b)\right)$$
$$\iff \Psi_g^{-1}(\theta_a) - \frac{\epsilon}{\sigma} \geq \frac{\epsilon}{\sigma} - \Psi_g^{-1}(1-\theta_b)$$
$$\iff \Psi_g^{-1}(\theta_a) + \Psi_g^{-1}(1-\theta_b) \geq \frac{2\epsilon}{\sigma}$$
$$\iff \Psi_g^{-1}(\theta_a) - \Psi_g^{-1}(\theta_b) \geq \frac{2\epsilon}{\sigma} \ .$$

### A.7.3 RELATION TO PIXELDP

*Pixel differential privacy* (pixelDP) was introduced in Lecuyer et al. (2019) using the same similarity measure between distributions used in differential privacy: a distribution-valued function $G : \mathbb{R}^d \to \mathcal{P}(\mathcal{Z})$ satisfies $(\varepsilon, \tau)$-*pixelDP with respect to $\ell_p$ perturbations* if for any $\|x - x'\|_p \leq 1$ it holds that $D_{\mathrm{DP},e^\varepsilon}(G(x)\|G(x')) \leq \tau$, where

$$D_{\mathrm{DP},e^\varepsilon}(G(x)\|G(x')) = \sup_E \left( \mathop{\mathbb{P}}_{X \sim G(x)}[X \in E] - e^\varepsilon \mathop{\mathbb{P}}_{X' \sim G(x')}[X' \in E] \right) \tag{19}$$

and the supremum is over all (measurable) subsets $E$ of $\mathcal{Z}$. In particular, Lecuyer et al. show that using a smoothing measure $\mu$ satisfying pixelDP with respect to $\ell^p$ leads to adversarially robust classifiers against $\ell^p$ perturbations.

To show that their result fits as a particular instance of our framework, take $\rho = \mu(x)$ and fix $\varepsilon \geq 0$ and $\tau \in [0, 1]$. Due to the symmetry of the constraint $\|x - x'\|_p \leq 1$, if $\mu$ satisfies $(\varepsilon, \tau)$-pixelDP with respect to $\ell_p$ perturbations, then we have the relaxation condition $\{\mu(x') : \|x - x'\|_p \leq 1\} \subseteq \mathcal{D}_{\varepsilon,\tau}$, where

$$\mathcal{D}_{\varepsilon,\tau} = \{\nu : D_{\mathrm{DP},e^\varepsilon}(\nu\|\rho) \leq \tau \text{ and } D_{\mathrm{DP},e^\varepsilon}(\rho\|\nu) \leq \tau\} \ . \tag{20}$$

Now we recall that Barthe & Olmedo (2013) noticed that $D_{\mathrm{DP},e^\varepsilon}$ is equivalent to the hockey-stick divergence $D_{\mathrm{HS},\beta}$ of order $\beta = e^\varepsilon$. Thus, since $f$-divergences are closed under reversal (property 4 in Appendix A.2), we see that the constraint set $\mathcal{D}_{\varepsilon,\tau}$ can be directly written in the form $\mathcal{D}_\mathcal{F}$ (cf. Section 2.2).

The main result in Lecuyer et al. (2019) is a limited-information black-box certification method for smoothed classifiers. The resulting certificate for, which provides certification with respect to $\mathcal{D}_{\varepsilon,\tau}$, is given by

$$\tau \leq \frac{\theta_a - e^{2\varepsilon}\theta_b}{e^\varepsilon + 1} \ . \tag{21}$$

For comparison, the certificate we obtain for the relaxation $\{\nu : D_{\mathrm{DP},e^\varepsilon}(\nu\|\rho) \leq \tau\}$ of $\mathcal{D}_{\varepsilon,\tau}$ (HS certificate in Table 4) already improves on the certificate by Lecuyer et al. whenever $\theta_a - \theta_b \geq (\beta - 1)(1 - \theta_a - \theta_b)$, which, e.g., always holds in the binary classification case. Furthermore, since Theorem 2 provides *optimal* certificates for $\mathcal{D}$, we have the following result.

**Corollary 7.** The optimal certificates for the constraint set $\mathcal{D}$ (cf. Eq. 20) obtained from Theorem 2 are stronger than those obtained from Eq. 21.

### A.8 EFFICIENT SAMPLING AND f-DIVERGENCE COMPUTATION FOR NORM-BASED SMOOTHING MEASURES

**Lemma 8.** The smoothing measure $\mu : \mathcal{X} \mapsto \mathcal{P}(\mathcal{X})$ with density $\mu(x)[z] \propto \exp(-\|z - x\|)$ satisfies

$$\max_{\|\delta\| \leq \epsilon} R_\infty(\mu(x + \delta)\|\mu(x)) \leq \epsilon \ .$$

if $\|x\|$ is any norm. Further, if $f$ is convex function with $f(1) = 0$ such that $f\left(\frac{1}{u}\right)$ is convex and monotonically increasing in $u$, then

$$\max_{\|\delta\| \leq \epsilon} D_f(\mu(x + \delta)\|\mu(x)) \leq \max_{\|\delta\| = \epsilon} \mathop{\mathrm{E}}_{X \sim \mu(0)}[f(\exp(-\|X - \delta\| + \|X\|))] \ . \tag{22}$$

*Proof.* By the triangle inequality, we have

$$\frac{\mu(x')[z]}{\mu(x)[z]} = \exp(\|z - x\| - \|z - x'\|) \leq \exp(\|x - x'\|)$$

so that

$$R_\infty(\mu(x')\|\mu(x)) \leq \|x - x'\| \ .$$

Similarly, for $f$ that satisfy the conditions of the theorem, it can be shown that $D_f(\mu(x')\|\mu(x))$ is convex in $x'$ so that its maximum over the convex set $\|x' - x\| \leq \epsilon$ is attained on the boundary. $\quad\square$

For several norms, the optimization problem in Eq. 22 can be solved in closed form. These include $\ell_1, \ell_2, \ell_\infty$ norms and the matrix spectral norm and nuclear norm (the final two are relevant when $\mathcal{X}$ is a space of matrices). The results are documented in Table 5. Thus, every $f$-divergence that meets the conditions of Lemma 8 can be estimated efficiently for these norms. In particular, the divergences that are induced by the functions $\tilde{f}(u^{-\alpha})$ for any monotonic convex function $\tilde{f}$ and $\alpha \geq 0$ satisfy this constraint. This gives us a very flexible class of $f$-divergences that can be efficiently estimated for these norm-based smoothing measures.

| Constraint on $\delta$ | Bound on Eq. 22 | Sampling from $X \sim \mu(0)$ |
|---|---|---|
| $\|\delta\|_1 \leq \epsilon$ | $\displaystyle \mathop{E}_{X \sim \mu_1(0)}[f(\exp(\|X - \epsilon e_0\|_1 - \|X\|_1))]$ | $X_i \sim \mathrm{Lap}(0,1)$ iid |
| $\|\delta\|_2 \leq \epsilon$ | $\displaystyle \mathop{E}_{X \sim \mu_2(0)}[f(\exp(\|X - \epsilon e_0\|_2 - \|X\|_2))]$ | $X = Ru$ 
 $R \sim \Gamma(d,1)$ 
 $u \sim \mathcal{U}(\partial \mathcal{B}_2)$ |
| $\|\delta\|_\infty \leq \epsilon$ | $\displaystyle \mathop{E}_{X \sim \mu_\infty(0)}[f(\exp(\|X - \epsilon\mathbf{1}\|_\infty - \|X\|_\infty))]$ | $X = Ru$ 
 $R \sim \Gamma(d+1,1)$ 
 $u \in \mathcal{U}(\mathcal{B}_\infty)$ |
| $\|\delta\|_{nuc} \leq \epsilon$ | $\displaystyle \mathop{E}_{\substack{s \sim \mu_1(0) \\ U,V \sim \mathcal{U}(\mathcal{O})}} \left[f\left(\exp\left(\left\|U[[s]]V^T - \epsilon[[e_0]]\right\|_{nuc} - \|s\|_1\right)\right)\right]$ | $X = U[[s]]V^T$ 

 $s \sim \mu_1, U, V \sim \mathcal{U}(\mathcal{O})$ |
| $\|\delta\|_\star \leq \epsilon$ | $\displaystyle \mathop{E}_{\substack{s \sim \mu_\infty(x) \\ U,V \sim \mathcal{U}(\mathcal{O})}} \left[f\left(\exp\left(\left\|U[[s]]V^T - \epsilon[[e_0]]\right\|_\star - \|s\|_\infty\right)\right)\right]$ | $X = U[[s]]V^T$ 

 $s \sim \mu_\infty, U, V \sim \mathcal{U}(\mathcal{O})$ |

Table 5: Bounds on $f$-divergences: $e_0$ is the vector with 1 in the first coordinate and zeros in all other coordinates and $\mathbf{1}$ is the vector with all coordinates equal to 1. $\mu_p$ refers to the smoothing measure induced by the $\ell_p$ norm, $\mathcal{U}(S)$ refers to the uniform measure over the set $S$, $\mathcal{O}$ is the set of orthogonal matrices and $\mathcal{B}_p = \{\|z\|_p \leq 1\}$ is the unit ball in the $\ell_p$ norm.

**Efficient sampling**  The only other requirement for obtaining a certificate computationally is to be able to sample from $\mu(x)$ to estimate $\theta_a, \theta_b$. Since $\mu(x)$ is log-concave, there are general purpose polynomial time algorithms for sampling from this measure. However, for most norms, more efficient methods exist, as outlined below.

The random variable $X \sim \mu(x)$ can be obtained as $X = x + Z$ with $Z \sim \mu(0)$. Thus, to sample from $\mu(x)$ for any $x$ it is enough to be able to sample from $\mu(0)$. For $\|\cdot\|_1$, this reduces to sampling from a Laplace distribution which can be done easily. For $\|\cdot\|_\infty$, (Steinke & Ullman, 2016) give the following efficient sampling procedure: first sample $r$ from a Gamma distribution with shape $d+1$ and mean $d+1$, i.e. $r \sim \Gamma(d+1,1)$, and then sample each $Z_i$, $i \in [d]$, uniformly from $[-r, r]$. Theorem 9 gives a short proof of correctness for this procedure. Theorem 10 also has a similar result for the case of $\|\cdot\|_2$ and Table 5 lists the sampling procedures for several norms.

**Theorem 9.** The random variable $Z \in \mathbb{R}^d$ obtained by first sampling $R \sim \Gamma(d+1,1)$ and then sampling each $Z_i$, $i \in [d]$, uniformly from $[-R, R]$ has density $\propto e^{-\|z\|_\infty}$.

*Proof.* We first compute the normalization constant for a density of the form $\propto e^{-\|z\|_\infty}$ as follows:

$$\int_{\mathbb{R}^d} e^{-\|z\|_\infty} dz = \int_0^\infty \left(\int_{\mathbb{R}^d} \mathbf{1}[\|z\|_\infty = t]dz\right)e^{-t}dt = \int_0^\infty 2^d dt^{d-1}e^{-t}dt = d!2^d \ .$$

Next we show the density of $Z$ satisfies $p_Z(z) = e^{-\|z\|_\infty}/(d!2^d)$ by noting that conditioned on $R = r$ we have $p_{Z|R=r}(z) = \mathbf{1}[\|z\|_\infty \leq r]/(2r)^d$ because of the uniform sampling used in each

coordinate, and integrating over $R$ sampled from a Gamma distribution with shape $d + 1$ and mean $d + 1$ yields

$$p_Z(z) = \int_0^\infty p_{Z|R=r}(z)p_R(r)dr = \int_0^\infty \frac{1[\|z\|_\infty \le r]}{(2r)^d}\frac{r^d e^{-r}}{d!}dr = \frac{1}{d!2^d}\int_{\|z\|_\infty}^\infty e^{-r}dr = \frac{e^{-\|z\|_\infty}}{d!2^d} \ .$$

$\square$

**Theorem 10.** The random variable $Z \in \mathbb{R}^d$ obtained by first sampling $Z' \sim \mathcal{N}(0, I)$ and $R \sim \Gamma_p(d, 1)$ and then taking $Z = R\frac{Z'}{\|Z'\|_2}$ has density $\propto e^{-\|z\|_2^p}$. Here $\Gamma_p(d, a)$ denotes the generalized Gamma distribution of order $p > 0$ with shape $d$ and scale $a$.

*Proof.* First note that $W = \frac{Z'}{\|Z'\|_2} \sim \mathcal{U}(\mathcal{B}_2)$; i.e. it is uniform on the $\ell_2$ ball of radius 1. Therefore, $RW$ is uniform on the $\ell_2$ ball of radius $R$ and the conditional density of $Z$ given $R$ is given by $p_{Z|R=r}(z) = \frac{1[\|z\|_2=r]\Gamma(d/2)}{2\pi^{d/2}r^{d-1}}$. Since $R$ has density $p_R(r) \propto r^{d-1}e^{-r^p}$, we get

$$p_Z(z) = \int_0^\infty p_{Z|R=r}(z)p_R(r)dr$$
$$\propto \int_0^\infty \frac{1[\|z\|_2 = r]\Gamma(d/2)}{2\pi^{d/2}r^{d-1}}r^{d-1}e^{-r^p}dr$$
$$\propto e^{-\|z\|_2^p} \ .$$

$\square$

## A.9 ALGORITHM FOR FULL-INFORMATION CERTIFICATION

In this section we describe the subroutines used in Algorithm 1. First we describe a generic procedure to provide high-probability confidence intervals for estimating expectations, then we give a detailed description of the subroutines.

### A.9.1 HIGH-CONFIDENCE ESTIMATES OF EXPECTED VALUES

Let $Z_1, \ldots, Z_N$ be independent, identically distributed random variables with range $R$ and mean $m$. Let the empirical mean be $\bar{Z} = \frac{1}{N}\sum_{i=1}^N Z_i$ and the empirical variance be $\bar{\sigma}^2 = \frac{1}{N}\sum_{i=1}^N(Z_i - \bar{Z})^2$. Applying Bernstein's inequality to the sum and the sum of the squares of these random variables, we get the empirical Bernstein bound (Audibert et al., 2009), which states that with probability at least $1 - \zeta$,

$$|\bar{Z} - m| \le \sqrt{\frac{2\bar{\sigma}^2\log(3/\zeta)}{N}} + \frac{3R\log(3/\zeta)}{t}. \tag{23}$$

The main benefit of the above inequality is that as long as the variance of the sample $Z_1, \ldots, Z_N$ is small, the convergence rate becomes essentially $O(1/N)$ instead of the standard $O(1/\sqrt{N})$. Also, since Eq. 23 only contains empirical quantities apart from the range $R$, it can be used to obtain computable bounds for the expectation $\mu$: with probability at least $1 - \zeta$,

$$\bar{Z} - \sqrt{\frac{2\bar{\sigma}^2\log(3/\zeta)}{N}} - \frac{3R\log(3/\zeta)}{N} \le m \le \bar{Z} + \sqrt{\frac{2\bar{\sigma}^2\log(3/\zeta)}{N}} + \frac{3R\log(3/\zeta)}{N} \ . \tag{24}$$

### A.9.2 SUBROUTINES FOR ALGORITHM 1

The bound in Eq. 24 can be applied to approximate the expectation in Eq. 6 with high probability for given values of $\lambda$ and $\kappa$. More specifically, if the function $f_\lambda^*(\kappa - \phi(\cdot))$ is bounded with range $R$, then taking $N$ samples $X_1, \ldots, X_N$ independently from $\rho$, and defining $Z_i = f_\lambda^*(\kappa - \phi(X_i))$, and $\bar{Z}$ and $\bar{\sigma}^2$ as above, Eq. 24 implies that with probability at least $1 - \zeta$,

$$\mathop{\mathrm{E}}_{X\sim\rho}[f_\lambda^*(\kappa - \phi(X_i))] \le \bar{Z} + \sqrt{\frac{2\bar{\sigma}^2\log(3/\zeta)}{N}} + \frac{3R\log(3/\zeta)}{N} \ . \tag{25}$$

Plugging in this bound to Eq. 6 gives a high-probability lower bound for the function to be maximized for any given $\lambda$ and $\kappa$.

Details of the above procedures are given in Algorithm 3. We use an off-the-shelf convex optimization solver (Diamond & Boyd, 2016) in the ESTIMETEOPT subroutine.

---

**Algorithm 3** Subroutines for Algorithm 1

---

    **function** ESTIMATEOPT$(\rho, \phi, N, \{f_i\}, \{\epsilon_i\})$
        Sample $X^1, \ldots, X^N \sim \rho$ and obtain $\kappa^*$ and $\lambda^*$ by solving Eq. 6 with $\mathbb{E}_{X \sim \rho}[f_\lambda^*(\kappa - \phi(X))]$ replaced by $\frac{1}{N} \sum_{i=1}^N f_\lambda^*\big(\kappa - \phi(X^i)\big)$ and the additional constraints $f_{\lambda^*}^*(\kappa - a) < \infty, f_{\lambda^*}^*(\kappa - b) < \infty$.
        **return** $\kappa^*, \lambda^*$.
    **end function**
    **function** UPPERCONFIDENCEBOUND$(\rho, \phi, \tilde{N}, \{f_i\}_{i=1}^M, \{\epsilon_i\}_{i=1}^M, a, b, \lambda, \kappa, \zeta)$
        Sample $X^1, \ldots, X^{\tilde{N}} \sim \rho$ and compute $Z^i \leftarrow f_\lambda^*\big(\kappa - \phi(X^i)\big)$ for $i = 1, \ldots, \tilde{N}$.
        Set $\bar{Z} \leftarrow \frac{\sum_{i=1}^{\tilde{N}} Z^i}{\tilde{N}}$.
        Set $\bar{\sigma} \leftarrow \frac{\sum_{i=1}^{\tilde{N}} (Z^i - \bar{Z})^2}{\tilde{N}}$.
        Set $R = \max_{x \in (a,b)} f_\lambda^*(\kappa - x) - \min_{x \in (a,b)} f_\lambda^*(\kappa - x)$.
        Set

$$E_{ub} \leftarrow \bar{Z} + \sqrt{\frac{2\bar{\sigma}^2 \log(3/\zeta)}{\tilde{N}}} + \frac{3R \log(3/\zeta)}{\tilde{N}} \,.$$

        **return** $E_{ub}$.
    **end function**

---

### A.10    $\ell_0$ SMOOTHING MEASURE

We can also handle discrete perturbations in our framework. A natural case to consider is $\ell_0$ perturbations. In this case, we assume that $\mathcal{X} = A^d$ where

$$A = \{1, \ldots, K\}$$

is a discrete set. Then, we can choose

$$\mu(x)[z] = \prod_{i=1}^d p^{1[z_i = x_i]} \left(\frac{q}{K-1}\right)^{1[z_i \neq x_i]} \tag{26}$$

where $p + q = 1$, $p \geq q \geq 0$, and $p$ denotes the probability that the measure retains the value of $x$ and $\frac{q}{K-1}$ denotes a uniform probability of switching it to a different value. In this case, it can be shown that for every $\alpha > 0$ that

$$R_\alpha(\mu(x')\|\mu(x)) = \|x - x'\|_0 \left( \frac{\log\left( p\left(\frac{q}{(K-1)p}\right)^{(\alpha)} + \frac{q}{K-1}\left(\frac{(K-1)p}{q}\right)^{(\alpha)} + \left(\frac{K-2}{K-1}\right)q \right)}{\alpha - 1} \right)$$

so that we can derive a certificate with respect to $\ell_0$ perturbations using any set of Rényi divergences (or combinations of theses).

This can be extended to structured discrete perturbations by introducing coupling terms between the perturbations:

$$\mu(x)[z] \propto \prod_{i=1}^d p^{1[z_i = x_i]} \left(\frac{q}{K-1}\right)^{1[z_i \neq x_i]} \exp\left( \sum_{i=1}^{d-1} \eta 1[z_i = x_i] 1[z_{i+1} = x_{i+1}] \right) \,.$$

This would correlate perturbations between adjacent features (which for example may be useful to model correlated perturbations for time series data). Since this can be viewed as a Markov Chain, Rényi divergences between $\mu(x), \mu(x')$ are still easy to compute.

A.11    COMPARISON WITH LECUYER ET AL. (2019) ON $\ell_1$ PERTURBATIONS

Here we compare our certificates to Lecuyer et al. (2019) on the MNIST, CIFAR-10 and ImageNet datasets. The smoothing distribution is as described in A.8; a zero mean Laplacian distribution with smoothing value defined by the scale of the distribution. We first describe the hyperparameters used in training and certification for each of the datasets. For all datasets, images were normalized into a [0,1] range.

**MNIST hyperparameters:**   We trained a standard three layer CNN ReLU classifier for 50,000 steps with a batch size of 128 and a learning rate of 0.001. The smoothing value during training was set to 1.0. For certification we use $N = 1K, \tilde{N} = 10M, \zeta = .99$, and sweep over a range of smoothing values between 0.5 and 1.5 and report the best certificate found. Certified accuracy is reported on 1,000 MNIST test set images.

**CIFAR-10 hyperparameters:**   We trained a Wide ResNet classifier for 50,000 training steps with a batch size of 32 and a learning rate of 0.001. The smoothing value during training was set to 0.2. For certification we use $N = 1K, \tilde{N} = 1M, \zeta = .99$, and sweep over a range of smoothing values between 0.1 and 0.5 and report the best certificate found. Certified accuracy is reported on 1,000 CIFAR-10 test set images.

**ImageNet hyperparameters:**   We trained a ResNet-152 classifier for 1 million training steps with a batch size of 16 and an initial learning rate of 0.1 that is decayed by a factor of ten every 25,000 steps. The smoothing value during training was set to 0.1. For certification we use $N = 1K, \tilde{N} = 100K, \zeta = .99$, and sweep over a range of smoothing values between 0.05 and 0.25 and report the best certificate found. Certified accuracy is reported on 500 ImageNet validation set images.

Table 6: $\ell_1$ comparison with Lecuyer et al. (2019) on MNIST.

| Certificate | Certified Accuracy | | | | | | |
|---|---|---|---|---|---|---|---|
| | $\epsilon = 1$ | $\epsilon = 2$ | $\epsilon = 3$ | $\epsilon = 4$ | $\epsilon = 5$ | $\epsilon = 6$ | $\epsilon = 7$ |
| Lecuyer et al. (2019) | 0.772 | 0.548 | 0.424 | 0.061 | 0 | 0 | 0 |
| Ours | 0.860 | 0.716 | 0.584 | 0.447 | 0.325 | 0.201 | 0.017 |

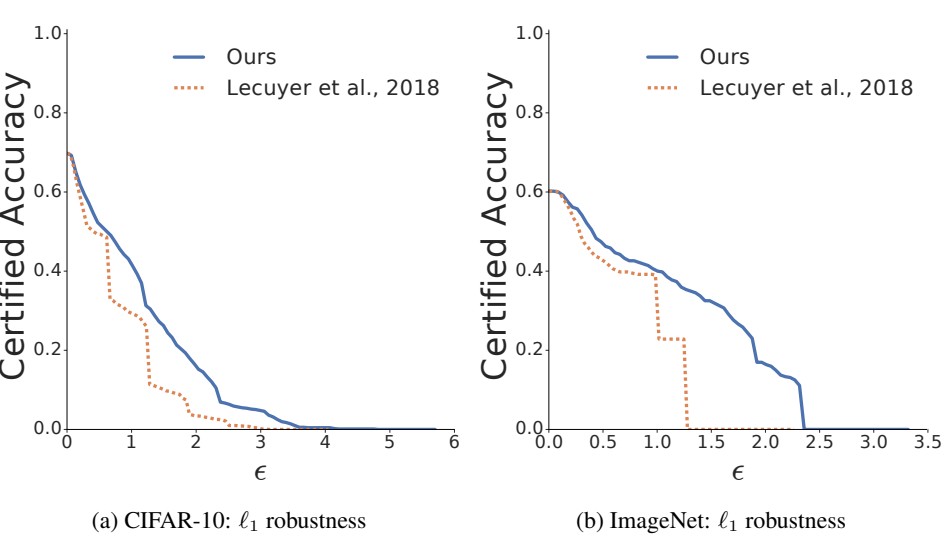

(a) CIFAR-10: $\ell_1$ robustness             (b) ImageNet: $\ell_1$ robustness

Figure 3: Certified accuracy under $\ell_1$ perturbations on CIFAR-10 and ImageNet.

Results for MNIST can be seen in Table 6, CIFAR-10 and ImageNet results are shown in Figure 3. We significantly outperform Lecuyer et al. (2019) on all three datasets.

