# OpenReview forum: "A FRAMEWORK  FOR ROBUSTNESS CERTIFICATION  OF SMOOTHED CLASSIFIERS USING  F-DIVERGENCES"
_ICLR.cc/2020/Conference — Accept (Poster)_

### Official Review · AnonReviewer1 · 2019-10-20
**Official Blind Review #1**

**Rating:** 6

**Review:**

The paper introduces a generalization of the randomized smoothing approach for certifying robustness of black-box classifiers, allowing the smoothing measure to be an arbitrary distribution (whereas previous work almost exclusively focused on Gaussian noise), and facilitating the certification with respect to different metrics under the same framework.

Given the wide interest in certified robustness based on randomized smoothing, the generalizations considered in this paper could have a high potential. I found the motivation, the definition of the framework and the statement of the main theoretical results, up to Section 3.2, very clear. The following sections were not as well organized, in my opinion, and need improvement. In particular, I found it not very clear how the framework can applied to recover previous results, in particular the results from Cohen et al. (2019). All the ingredients seem to be there, among Theorem 3, the result from Balle & Wang (2018), and Corollary 5, but the arguments could be presented in a better organized way.

What I also found is missing is a clear descriptions of practical algorithms for applying the framework. Again, most of the ingredients seem to be there (e.g. Table 1 in the main body, Lemma 6 as well as the discussion about sampling mechanism in the Appendix), but they are not well organized. Presumably, the same statistical methodology as in Cohen et al. is used to obtain estimates of theta_a and theta_b, but this doesn't seem to be clearly stated anywhere. How is certification practically performed for intersections of contraint sets as introduced at the end of section 2? How the full-information certification can be applied based on empirical estimates is unclear to me, too.

Finally, the description of the experiments need improvements: For instance, in Section 5.1, which smoothing measure was used? Was the ResNet classifier trained using samples from this measure? Which sets of f-divergences were used? What does Figure 3 (a) exactly show? Do the blue points correspond to the 50 data samples? The number of random samples for computing empirical estimates and the confidence bounds are missing.
In Figure 4 (a), as the l_0 norm counts the number of altered pixels I don't understand why the certified accuracy varies e.g. for epsilon *between* 0 and 1.
The experiment on the Librispeech model seems interesting, but the paper does not contain sufficient information to understand and assess the experimental set-up or the results.

In summary, while I believe that the generalizations proposed in this paper have potential, in its present form the manuscript doesn't describe clearly and accurately enough how the framework can be applied to recover previous theoretical results or perform certification in practice.

----------

I acknowledge I have read the authors' response. As it addresses my main concerns I have changed my rating from "weak reject" to "weak accept".


**Experience Assessment:**

I have read many papers in this area.

**Review Assessment: Checking Correctness Of Derivations And Theory:**

I assessed the sensibility of the derivations and theory.

**Review Assessment: Checking Correctness Of Experiments:**

I assessed the sensibility of the experiments.

**Review Assessment: Thoroughness In Paper Reading:**

I read the paper at least twice and used my best judgement in assessing the paper.

---

> ### Author Response · Authors · 2019-11-12
> **Thank you for your comments**
>
> We thank the reviewer for the careful review and encouraging feedback and hope that our updated draft and the following clarifications address the concerns raised by the reviewer.
>
> Clarification on connection to Cohen et al: We have significantly revised Section 5 (Connections to prior work) and clarified the steps involved in deriving the Cohen et al result as a special case of our framework. We refer to the paragraph titled "Cohen et al 2019" below table 1 in the revised draft.
>
> Clarification on practical certification algorithms: We have revised section 3 of the paper
> so that theorems 1 and 2 are immediately followed by descriptions of practical certification algorithms that derive from these theorems - we refer the reviewer to algorithm 2 (for the information-limited case) and algorithm 1 (for the full-information case). We have also added section 2.3 on obtaining bounds on f-divergences.
>
> Clarification on experiments:
> 1. Full-information vs Information limited setting:  We have revised definitions 2.1 and 2.2 to better reflect this distinction. We added section 4.2 where we prove that the full-information certificates are provably tighter than the information-limited ones when smoothing probabilistic classifiers. We have added section 6.1 in the experiments that clarifies the details of the experiment comparing the full-information setting with the limited-information setting for l2 perturbations.
> 2. L0 certification: To address the issue of fractional l0 values in the plot, we have removed the plots and produced tables that document the certified robustness as a function of the radius of the l0 perturbation (tables 2 and 3).
> 3. Librispeech: We have expanded the experiments section to add more details about the Librispeech dataset as well as motivation for performing these experiments in section 6.3. We are happy to answer any further questions on this.

---

### Official Review · AnonReviewer3 · 2019-10-21
**Official Blind Review #3**

**Rating:** 6

**Review:**

Summary: This submission proposes a unified framework for black-box adversarial certification. Based on cohen's and lee's result, it extends the \lambda_TV distance to more general f-divergence. Besides, some other techniques, e.g. reference distribution, are also included in the framework. For L_0 and L_1 perturbation, the proposed framework comes to better results than all the related methods.

Strengths:
[+] Detailed analysis and theorem.
[+] Describing the background and conclusions clearly.

Weaknesses:
[-] In the experiment section, for ImageNet, baselines use ResNet-50 but the authors use ResNet-152. I wonder whether this is fair enough.
[-] Renyi divergence is not a proper f-divergence. Therefore, maybe the title should be changed.
[-] The theorem part contains too many details and some important parts (e.g. tightness) are in appendix. It should be re-organized.

Questions:
[.] In Appendix A.7, Lagrangian strong-duality has been used to show the tightness, but I wonder whether the tightness holds for any kind f-divergence? If the authors can write down some theorem about the tightness, it will be better.

p.s. Could you give me some quick responses to my questions (e.g. ResNet-152 v.s. ResNet-50) ? Once you convince me, I would revise the score to 6. If not, I will turn down the score to 3.

**Experience Assessment:**

I have read many papers in this area.

**Review Assessment: Checking Correctness Of Derivations And Theory:**

I assessed the sensibility of the derivations and theory.

**Review Assessment: Checking Correctness Of Experiments:**

I assessed the sensibility of the experiments.

**Review Assessment: Thoroughness In Paper Reading:**

I read the paper at least twice and used my best judgement in assessing the paper.

---

> ### Author Response · Authors · 2019-11-12
> **Thank you for your comments**
>
> We thank the reviewer for the careful review and encouraging feedback. We address the comments of the reviewer below:
> 1. We have re-run the experiments with ResNet-50 for ImageNet. We took the model trained by Lee et al and ran comparisons between our certification procedure and those from Lee et al, the results are shown in Table 2.
> 2. Yes, the Renyi divergence is not an f-divergence but is related via 1-1 mapping to an f-divergence. We have further reflected this in the paper see Section 2.2, paragraph titled “Relaxations using f-divergence”. Further, we have theoretical results (theorem 4) and experimental results (section 6.1) that use the hockey-stick divergence, and hence we think f-divergences is an appropriate unifying title.
> 3. We have reorganized sections 3 and 4 in the paper and now have a full statement of the tightness result in theorem 4 in the paper.
> 4. As shown in theorem 4, the tightness result in the information-limited holds specifically for the case of hockey-stick divergences. However, since it is not easy to compute hockey-stick divergences in general (Gaussians being a notable exception), it is also valuable to have certification procedures that use other f-divergences. We are happy to provide any further clarifications the reviewer seeks in this regard.

---

> ### Author Response · Authors · 2019-11-13
> **p.s. clarification and score revision**
>
> We would like to follow up on the p.s request from the reviewer - we believe that this has been addressed in our revision (section 6.2, table 2) and are happy to provide any further clarifications the reviewer seeks.

---

### Official Review · AnonReviewer2 · 2019-10-22
**Official Blind Review #2**

**Rating:** 6

**Review:**

This paper extends existing work on certified robustness using smoothed classifier.
The fundamental contribution is a framework that allows for arbitrary smoothing measure, in which certificates can be obtained by convex optimization.
A good number of technical contributions
Framework for certificate under arbitrary smoothing measure -> Theorem 1, and proof in A.4 (good use of duality)
Full calculation/result of certificates under different divergneces in Table 1.
Reasonable set of empirical evidence.

Overall a lot of good things to be said, below are some questions/comments that could improve the paper:
*Technical*
Personally, I cannot get a lot of value out of the distinction between full-information and information-limited certification.  It’d be great if I can get some clarification on this.
*Experiments*
Generally, more details of the experiments should be included.
How are the convex optimization problems actually solved (e.g., what methods/tools)?
How much more computational overhead is there?
Oddly, seems like we’re missing CIFAR10 results completely.

Sec 5.1.,
What does each dot in Figure 3a represent?
Sec 5.2,
Somewhat strangely, Figure 3b is results on l0 perturbation, but not l2.  What happens for l2 when we use M>1?  How does this compare to other extensions (likely the SOTA) like [1]?
Sec 5.3,
It is unclear from the writing whether the comparisons to previous works were done on the same ResNet architecutre with the same clean accuracy.  Please clarify.
I’m not sure why we need the Librispeech results.  It’s not motivated clearly.  Also, from writing it seems the adversary zeros out a segment.  It’s unclear if this is a reasonable kind of attack to expect on speech.  If I block out a segment of the speaker, we probably don’t expect any system to do well on speaker recognition.  I suggest removing this result, or somehow make it a lot more better motivated/conducted.  Clarify if I missed reasons why simply showing your method works on speech is impressive.

Here are suggestions on writing:
Contribution --- in both the abstract and introduction, the experimental results should be stated clearer.  Be more specific, e.g., “Show SOTA certified l2 robustness on X,Y,Z, establish first certified robustness on Librispeech, and first results on certified l0, l1 robustness on A,B,C.”
More broadly in the introduction, please motivate why “adversarial attacks as measured by other smoothing measure is important”.  Past studies focus on l2-norm not just because they are do-able, but also white noise (which is naturally measured by l2-norm) is something to expect in practice.  Justify why l0, l1 would also be important.

I recommend accepting this paper, but would do so with more passion if some of the comments/questions can be addressed.

Best,

Reference:
[1] https://arxiv.org/abs/1906.04584


**Experience Assessment:**

I do not know much about this area.

**Review Assessment: Checking Correctness Of Derivations And Theory:**

I assessed the sensibility of the derivations and theory.

**Review Assessment: Checking Correctness Of Experiments:**

I carefully checked the experiments.

**Review Assessment: Thoroughness In Paper Reading:**

I read the paper thoroughly.

---

> ### Author Response · Authors · 2019-11-12
> **Thank you for your comments**
>
> We thank the reviewer for the careful review and encouraging feedback. We respond to individual comments below:
> Technical
> On the distinction between information-limited and full-information certificates: We have updated the paper to clarify this distinction and refer the reviewer to section 2 (definitions 2.1 and 2.2) - we have updated the definitions. Further, section 4.1 shows theoretically how the full-information setting improves upon the information-limited setting and section 6.1 verifies this experimentally.
>
> Experiments: We have added more details to experiments (section 6 in the updated draft). The remaining comments are address below:
> 1. Solution of convex optimization problems: We use an off-the-shelf solver CVXPY to solve the convex optimization problems for the experiments in section 6.1. We have documented the certification running times in section 6.1 for this procedure as well (see caption of Figure 2). In the information-limited setting considered in sections 6.2 and 6.3, we can simply use the closed-form certificates computed in the table 4 (appendix section A.6).
> 2. CIFAR-10: We have added results for l1 certification on CIFAR-10 in appendix section A11.
> 3. Sec 5.1:We thank the reviewer for pointing this out, we have updated the paper to reflect this (it is now Fig. 2 in section 6.1). Please refer to the updated draft.
> 4. Sec 5.2: For L2 perturbations, the optimal certificate in the information limited setting can be obtained from Cohen et al, which is a special case of our framework that uses M=2 Hockey-Stick divergences (section 4, Corollary 5). We have removed this figure due to space constraints in the revised draft.
> 5. Comparison to [1]: The contribution of our paper is a general flexible framework for certifying robustness of smoothed classifiers while [1] focuses on using the certification method from (Cohen et al 2019) but improving the training so as to obtain classifiers with high certified accuracy. Thus, the contributions of our paper and [1] can be considered to be orthogonal, and indeed combining them is an interesting future research direction.
> 6. Sec 5.3: We have revised these results so that the model used for both certification procedures is the same and thus the clean accuracy is also the same. We have further clarified this in the paper (see Table 2, section 6.2).
> 7. Motivation behind librispeech experiments: We have added clarifying text in section 6.3 in the revised draft on the motivation.
>
> Comments on writing:
> We thank the reviewer for these suggestions and have incorporated them into the revised text. We have revised the abstract and contributions sections in the paper to make the specific contributions of the paper clear. The contributions section also now contains justification for the perturbations studied in the paper.

---

### Public Comment · ~Bai_Li1 · 2019-11-05
**Closely Related Previous Work**

Dear Authors,

Thank you for the interesting work. I would like to point out that using Renyi divergence to calculate certified bound for randomized smoothing has been extensively studied in https://arxiv.org/abs/1809.03113 [1].

Although this paper aims to generalize the framework to general f-divergence, Renyi divergence is particularly used in the experiments, which is the exact focus of [1]. The comparison of Renyi divergence based bound and the one obtained from PixelDP has also been done in [1].

I suggest the authors discuss the connection and distinction between their work and [1] for the comprehensiveness.

[1] Li, B., Chen, C., Wang, W., & Carin, L. (2018). Certified Adversarial Robustness with Additive Noise. arXiv preprint arXiv:1809.03113.

---

> ### Author Response · Authors · 2019-11-08
> **Thanks for pointing this out**
>
> Dear Bai,
> Thank you for bringing this work to our attention.
>
> We agree that the Renyi divergence case is indeed closely related and we are preparing an updated draft clarifying the distinction and relationship with [1].
>
> In particular, we believe that our work generalizes [1] in three ways:
> 1. Our framework handles arbitrary smoothing measures (not just Gaussian and Laplacian).
> 2. We allow arbitrary f-divergence measures. In particular, for certifying l2 perturbations using Gaussian smoothing, we show that the hockey-stick divergences are necessary to obtain optimal certificates (recovering the work of Cohen et al).
> 3. Even in the specific case of Renyi divergences, we can obtain better certificates by using convex combinations of Renyi divergences.
>
> We will update the draft to reflect this and are happy to answer any other questions.
>
> Regards,
> Authors

---

### Public Comment · ~Guang-He_Lee1 · 2019-11-08
**Some comments**

Dear Authors,

Thank you for the effort in preparing this work, but we have some questions and concerns about the paper.

Taking numbers directly from another paper [1] for comparison is appropriate only if the protocol was the same. This doesn’t seem to be the case. For example, [1] used 0.999 confidence level with 100K samples, while in this paper confidence level is 0.99 with 10M samples. So one could get improvements even with exactly the same certification algorithm. Also, since the difference is really the certification algorithms rather than models, are the models the same?

Besides, we would like to clarify your descriptions of [1]. For instance, “they require an O(d^3) computation (where d is the input dimension) to certify smoothness to L0 perturbations (in addition to the cost of estimating θa, θb by sampling).” The O(d^3) computation only requires to be done once for any given dataset. After the computation is done once, the certification takes constant time for any input data and model for the dataset: testing whether θa > some pre-computed threshold.

The description “Finally, these works are both restricted to the information-limited black-box verification setting where only θa, θb are known.” is incorrect. To our best knowledge, [1] is the first one to extend robustness certificates for randomized smoothing beyond only knowing θa, θb. For example, given a decision tree predictor under the smoothed distribution, [1] showed that the exact certificate can be efficiently computed.

Finally, could you please explain how (θa, θb) are estimated in the experiments? For each image, did you use the same 10M samples for estimating θa and θb?

[1] “Tight Certificates of Adversarial Robustness for Randomly Smoothed Classifiers.” (The old title is “A Stratified Approach to Robustness for Randomly Smoothed Classifiers”), NeurIPS 2019.

---

> ### Author Response · Authors · 2019-11-08
> **Thank you for the clarification**
>
> Dear Guang-He,
>
> Thank you for taking time to read our paper and pointing out these concerns.
>
> The goal of our paper is to develop a general framework for certifying smoothed classifiers that allows arbitrary smoothing measures and perturbations. Indeed, in the case of l0 perturbations, the method from [1] is optimal. We will clarify this in our updated draft and change the experiments to better reflect this.
>
> More concretely, we plan to make the following changes:
>
> 1. Experiments: Regarding the experimental protocol, you are correct in pointing out the differences. We will rectify this by rerunning our experiments with a consistent experimental protocol, and if possible within the time constraints of the rebuttal period, with the same models (since our code is in tensorflow, it may take some time to convert the pytorch models and rerun the experiments).
>
> 2. We agree that the O(d^3) computation is a one-time computation for a given smoothing measure and dataset. We will clarify this in the text. We would like to point that the quicker computation time in our approach allowed us to experiment with a broader range of smoothing measures which indeed can improve model performance significantly.
>
> 3. We will change the sentence “Finally, these works are both restricted to the information-limited black-box verification setting where only θa, θb are known.” to clarify that [1] indeed looked at using more information to obtain better certificates.  We would like to point out that our full-information certificates are still black-box (and hence apply to arbitrary classifiers including deep nets) and do not require access to the internals of the classifier, unlike the decision tree results obtained in [1].
>
> 4.  (θa, θb) are estimated using the same procedure as Cohen et al (the Clopper-Pearson CI test). We use 10M samples for MNIST, 1M samples for CIFAR-10 and 100K samples for ImageNet - these are sampled independently for every image.

---

### Author Response · Authors · 2019-11-12
**Summary of rebuttal**

We thank the blind reviewers as well as authors of the public comments for their careful reviews and feedback on our paper. We have made the following changes to our paper in response:
1. We have organized the technical material in the paper into three sections:
       Section 3: The main certification theorems, followed by descriptions of practical certification algorithms that follow from the theorems
       Section 4: Theoretical analysis of the certification algorithms
       Section 5: Comparisons to prior work, including a table comparing the contributions of our work relative to prior work (table 1) and detailed explanations relating our work to prior work
2. We have amended the abstract and contributions sections of our paper to better reflect our contributions relative to prior work and motivate the use of the perturbation measures (l0, l1 and l2) used.
3. We have rerun experiments on l0 certification to ensure an experimental protocol that is consistent with prior work ensuring fair comparisons can be made. We have expanded the descriptions in the experiments section to clarify precise details of our experimental protocol and highlight novel contributions from our work.

---

### Decision · Program_Chairs · 2019-12-19

**Decision:**

Accept (Poster)

**Comment:**

This submission proposes a black-box method for certifying the robustness of smoothed classifiers in the presence of adversarial perturbations. This work goes beyond previous works in certifying robustness for arbitrary smoothing measures.

Strengths:
-Sound formulation and theoretical justification to tackle an important problem.

Weaknesses
-Experimental comparison was at times not fair.
-The presentation and writing could be improved.

These two weaknesses were sufficiently addressed during the discussion. All reviewers recommend acceptance.